# Mitochondrial origins of the pressure to sleep

Raffaele Sarnataro[1], Cecilia D. Velasco[1], Nicholas Monaco[1], Anissa Kempf[1,2] & Gero Miesenböck[1✉]

To gain a comprehensive, unbiased perspective on molecular changes in the brain that may underlie the need for sleep, we have characterized the transcriptomes of single cells isolated from rested and sleep-deprived flies. Here we report that transcripts upregulated after sleep deprivation, in sleep-control neurons projecting to the dorsal fan-shaped body[1,2] (dFBNs) but not ubiquitously in the brain, encode almost exclusively proteins with roles in mitochondrial respiration and ATP synthesis. These gene expression changes are accompanied by mitochondrial fragmentation, enhanced mitophagy and an increase in the number of contacts between mitochondria and the endoplasmic reticulum, creating conduits[3,4] for the replenishment of peroxidized lipids[5]. The morphological changes are reversible after recovery sleep and blunted by the installation of an electron overflow[6,7] in the respiratory chain. Inducing or preventing mitochondrial fission or fusion[8–13] in dFBNs alters sleep and the electrical properties of sleep-control cells in opposite directions: hyperfused mitochondria increase, whereas fragmented mitochondria decrease, neuronal excitability and sleep. ATP concentrations in dFBNs rise after enforced waking because of diminished ATP consumption during the arousal-mediated inhibition of these neurons[14], which augments their mitochondrial electron leak[7]. Consistent with this view, uncoupling electron flux from ATP synthesis[15] relieves the pressure to sleep, while exacerbating mismatches between electron supply and ATP demand (by powering ATP synthesis with a light-driven proton pump[16]) precipitates sleep. Sleep, like ageing[17,18], may be an inescapable consequence of aerobic metabolism.

Sleep pressure, the process variable in sleep homeostasis, has lacked a physical interpretation. Although prolonged waking is associated with numerous changes in the brain—of neuronal firing patterns[19,20], the strengths of synaptic connections[21], the organization of subcellular compartments[22–24], metabolite concentrations[25,26] and metabolic and gene expression programs[23,27,28]—it remains generally indeterminable whether these changes are causes or consequences of a growing need for sleep. Perhaps the only opportunity for separating causation from correlation exists in specialist neurons with active roles in the induction and maintenance of sleep[29]; in these cells, sleep's proximate (and maybe also its ultimate) causes must interlock directly with the processes that regulate spiking. To delineate the molecular determinants of these processes in as complete and unbiased a manner as possible, we collected single-cell transcriptomes[30] of the brains of rested and sleep-deprived flies (Extended Data Fig. 1a). An encodable fluorescent marker allowed us to identify and enrich for two dozen sleep-inducing neurons with projections to the dorsal fan-shaped body of the central complex[1,2] (dFBNs) and compare their transcriptomic response to sleep loss with that of other identifiable cell types.

## Traces of sleep loss in mitochondria and synapses

*Drosophila* brains were dissociated into single-cell suspensions, and neurons expressing GFP under the control of *R23E10-GAL4* (ref. 31) were isolated by flow cytometry (Extended Data Fig. 1b). We performed single-cell RNA sequencing (scRNA-seq; 10X Chromium) on cells contained in the GFP-positive and GFP-negative fractions; among the 13,173 high-quality cells retrieved (Extended Data Fig. 1c,d), neurons nominated by *R23E10-GAL4* were identified by the expression of *GFP*, the neuronal markers *embryonic lethal abnormal vision* (*elav*) and *neuronal synaptobrevin* (*nSyb*), and *Allatostatin A* (*AstA*) *receptor 1* (*AstA-R1*), whose transcriptional enhancer provides the genomic fragment driving expression in the *R23E10-GAL4* line. Most neurons in the *R23E10-GAL4* domain contained machinery for the synthesis and release of either glutamate or GABA (γ-aminobutyric acid) (Extended Data Fig. 1e); some also transcribed the gene encoding the vesicular acetylcholine transporter (VAChT) (Extended Data Fig. 1e)—a common occurrence in non-cholinergic neurons, which rely on a microRNA to suppress its translation[32]. Indeed, fluorescent sensors detect the release of glutamate, but not of acetylcholine, from dFBNs[2].

In the *Drosophila* CNS, glutamate and GABA both act on ligand-gated chloride channels, consistent with the known inhibitory effect of dFBNs on their postsynaptic partners[2,33]. Members of the *R23E10-GAL4* pattern that signal via one or the other of these inhibitory transmitters are anatomically segregated[2]: a pair of GABAergic cells target the suboesophageal ganglion, while a larger glutamatergic population innervates the dorsal fan-shaped body. This latter group of bona fide dFBNs formed a distinct gene expression cluster of 323 cells among all glutamatergic neurons in the brain (Fig. 1a and Extended Data Fig. 1f,g). The cluster was defined by genes controlled by known dFBN enhancers

[1]Centre for Neural Circuits and Behaviour, University of Oxford, Oxford, UK. [2]Present address: Biozentrum, Universität Basel, Basel, Switzerland. ✉e-mail: gero.miesenboeck@cncb.ox.ac.uk

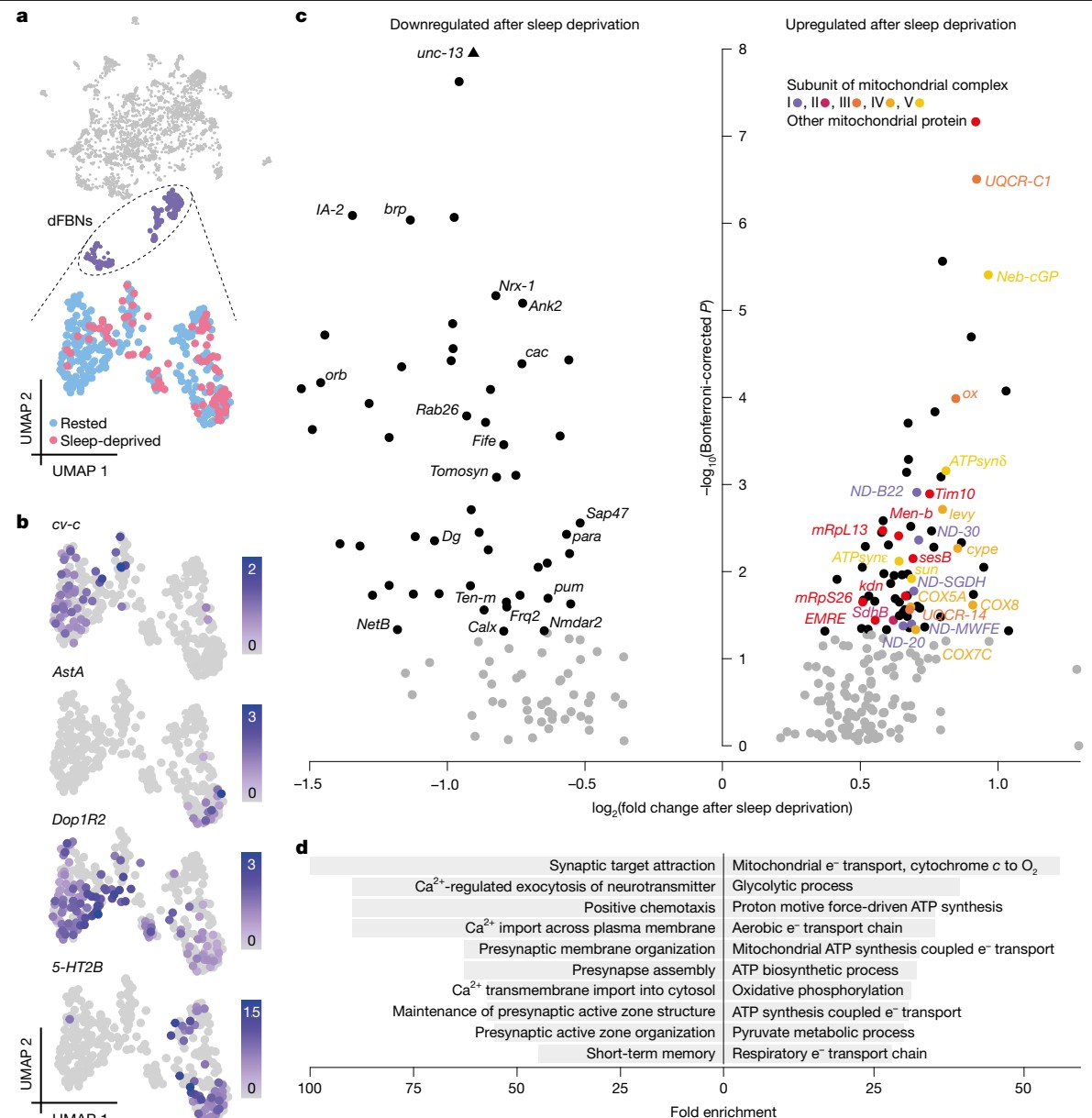

**Fig. 1 | The transcriptional response of dFBNs to sleep deprivation. a**, Uniform manifold approximation and projection (UMAP) representation of glutamatergic neurons (grey) according to their gene expression profiles. dFBNs (purple) form a distinct cluster containing cells from rested (blue, $n = 237$ cells) and sleep-deprived brains (red, $n = 86$ cells). **b**, log-normalized expression levels of dFBN markers. **c**, Volcano plot of sleep history-dependent gene expression changes in dFBNs. Signals with Bonferroni-corrected $P < 0.05$ (two-sided Wilcoxon rank-sum test) are indicated in black; labels identify protein products localized to synapses or mitochondria; colours denote subunits of mitochondrial respiratory complexes. The $P$ value of *unc-13* exceeds the $y$-axis limit and is plotted at the top of the graph. **d**, Enrichment of the top ten downregulated (left) and upregulated (right) 'biological process' gene ontology terms in the set of differentially expressed dFBN genes.

(Extended Data Fig. 1h) and encoding known dFBN markers, such as the Rho GTPase-activating protein[31] crossveinless-c, the dopamine receptor[14] Dop1R2, the serotonin receptor[34] 5-HT2B and the neuropeptide[33] AstA (Fig. 1b), but the distributions of some markers suggested further subdivisions. For example, AstA was confined to a subset of dFBNs (Fig. 1b), consistent with the comparatively mild sleep phenotype following RNA-mediated interference (RNAi) with its expression in the entire dFBN population[33].

Gene ontology analyses of the 122 transcripts whose levels in dFBNs changed after 12 h of overnight sleep deprivation pointed to only two sleep need-dependent processes: mitochondrial energy metabolism and synaptic transmission (Fig. 1c,d and Extended Data Fig. 2a–e). Sleep loss led to the selective upregulation of transcripts encoding

components of electron transport complexes I–IV, ATP synthase (complex V), the ATP–ADP carrier sesB and enzymes of the tricarboxylic acid cycle (the citrate synthase kdn, the B subunit of succinate dehydrogenase and the malate dehydrogenase Men-b), whereas gene products involved in synapse assembly, synaptic vesicle release and presynaptic homeostatic plasticity[35] were selectively downregulated (Fig. 1c). Within the practical limits of our analysis, which cannot extend to every conceivable neuron type, this transcriptomic signature of sleep loss appeared unique to dFBNs: it was absent from two cell populations with comparable numerical representation in our data, namely projection neurons of the antennal lobe (317 cells) (Extended Data Fig. 3a–d) and Kenyon cells of the mushroom body (603 cells) (Extended Data Fig. 3e–h), and it was also undetectable in a combined analysis of all

12,850 non-dFBN cells (Extended Data Fig. 3i,j). There are, however, subtle hints from an independent transcriptomic study that sleep history may alter the levels of mitochondrial components not only in dFBNs but also in R5 neurons of the ellipsoid body, another element of the sleep homeostat[28].

The remainder of this Article examines the causes and consequences of the differential expression of genes encoding mitochondrial proteins in dFBNs; a parallel study[2] investigates the role of presynaptic plasticity in the wider context of sleep need-dependent dFBN dynamics.

## A mitochondrial electron surplus induces sleep

The prominence of mitochondrial components in the transcriptional response of dFBNs to sleep deprivation (Fig. 1c,d and Extended Data Fig. 2) offers unbiased support for the hypothesis that sleep and aerobic metabolism are fundamentally connected. This hypothesis gained a firm mechanistic footing with the discovery that dFBNs regulate sleep through machinery that gears their sleep-inducing spike discharge to mitochondrial respiration[7]. The centrepiece of this mechanism is Hyperkinetic, the β-subunit of the voltage-gated potassium channel Shaker, which regulates the electrical activity of dFBNs[7,14]. Hyperkinetic is an unusual aldo-keto reductase with a stably bound nicotinamide adenine dinucleotide phosphate cofactor whose oxidation state (NADPH or NADP[+]) reflects the fate of electrons entering the respiratory chain[5,7]. When the demands of ATP synthesis are high, the vast majority of electrons reach $O_2$ in an enzymatic reaction catalysed by cytochrome c oxidase (complex IV); only a small minority leak prematurely from the upstream mobile carrier coenzyme Q (CoQ), producing superoxide and other reactive oxygen species[17,36] (ROS) (Fig. 2a). The probability of these non-enzymatic single-electron reductions of $O_2$ increases sharply under conditions that overfill the CoQ pool as a consequence of increased supply (high NADH to NAD[+] ratio) or reduced demand (large protonmotive force ($\Delta p$) and high ATP to ADP ratio)[17,18,36] (Fig. 2a). The mitochondria of dFBNs are prone to this mode of operation during waking[7], when caloric intake is high but the neurons' electrical activity is reduced, leaving their ATP reserves full. Indeed, measurements with the genetically encoded ATP sensors iATPSnFR and ATeam showed approximately 1.2-fold higher ATP concentrations in dFBNs, but not projection neurons, after a night of sleep deprivation than at rest (Fig. 2b,c and Extended Data Fig. 4a,b). ATP concentrations rose acutely when dFBNs were inhibited by an arousing heat stimulus, which releases dopamine onto their dendrites[2,14] (Fig. 2a,d), and fell below baseline when dFBNs themselves were stimulated, mimicking sleep[2] (Fig. 2a,e).

Even if the chance of an individual electron spilling from the CoQ pool is low, however, metabolically highly active cells, such as neurons, will by the sheer number of electrons passing through their respiratory chains generate significant amounts of ROS[17,18,36,37]. Their anti-cyclical relationship between energy availability (which peaks during waking) and energy consumption (which in a sleep-active neuron peaks during sleep) may thus predispose dFBNs to an exaggerated form of the electron leak experienced by many neurons in the awake state, making them an effective early warning system against widespread damage. Because polyunsaturated membrane lipids are especially at risk[5], dFBNs estimate the size of the mitochondrial electron leak indirectly, by counting reductions of lipid peroxidation-derived carbonyls at Hyperkinetic's active site[5].

Several lines of evidence point to a mismatch between the number of electrons entering the mitochondrial transport chain and the number needed to fuel ATP production as a root cause of sleep. First, opening an exit route for surplus electrons from the CoQ pool (by equipping the mitochondria of dFBNs with the alternative oxidase[6] (AOX) of *Ciona intestinalis*, which produces water in a controlled four-electron reduction of $O_2$) not only relieved the basal pressure to sleep[7] but also remedied the excessive sleep need of flies whose ability to remove breakdown products of peroxidized lipids was impaired[5]. Second,

increasing the demand of dFBNs for electrons (by overexpressing the uncoupling proteins Ucp4A or Ucp4C, which short-circuit the proton electrochemical gradient across the inner mitochondrial membrane (IMM)[15]) (Fig. 2a) decreased sleep (Fig. 2f and Extended Data Fig. 4c). And third, powering ATP synthesis with photons rather than electrons (by illuminating a mitochondrially targeted version of the light-driven archaeal proton pump delta-rhodopsin[16]) (Fig. 2a and Extended Data Fig. 4d) made NADH-derived electrons in dFBNs redundant and precipitated sleep (Fig. 2g,h and Extended Data Fig. 4e,f).

## Sleep alters mitochondrial dynamics

Given this wealth of evidence, it is not surprising that mitochondria would emerge as one of two pivots in the reorganization of dFBN gene expression after sleep deprivation (Fig. 1c,d and Extended Data Fig. 2). However, it remains ambiguous whether the upregulation of transcripts encoding mitochondrial proteins signals a net increase in mitochondrial mass or a compensatory response to organelle damage. To disambiguate these scenarios, we labelled the mitochondria of dFBNs with matrix-localized GFP (mito-GFP) and, in independent series of experiments, imaged the neurons' dendritic fields by confocal laser-scanning microscopy (CLSM) or optical photon reassignment microscopy[38] (OPRM). Both forms of light microscopy were validated against ultrastructure: automated morphometry of deconvolved image stacks established that, although both optical methods overestimated the true organelle size to some extent, the size ratios of mitochondria in dFBNs and uniglomerular olfactory projection neurons of the antennal lobes matched that determined by volume electron microscopy[39] (Extended Data Fig. 5a,b). Although diffraction undoubtedly inflated the absolute dimensions of mitochondria, especially in CLSM images, valid inferences about relative morphological differences could therefore still be drawn.

A night of sleep loss, regardless of whether caused by mechanical agitation or an artificial elevation of arousing dopamine[14], reduced the size, elongation and branching of dFBN mitochondria (Fig. 3a,b and Extended Data Fig. 5c–f) and led to the relocation of dynamin-related protein 1 (Drp1), the key fission dynamin of the outer membrane[8–10], from the cytosol to the mitochondrial surface (Fig. 3c). OPRM, which achieves lateral super-resolution approximately twofold above the diffraction limit[38], detected a concomitant increase in mitochondrial number (Fig. 3b), indicative of organelle division, which eluded CLSM (Extended Data Fig. 5f, but see Methods for a cautionary remark about the interpretation of absolute mitochondrial counts). This seemingly incomplete tally of daughter mitochondria notwithstanding, the two methods painted congruous pictures of mitochondrial fission after sleep loss (Fig. 3a,b and Extended Data Fig. 5c–f). The mitochondria of antennal lobe projection neurons, by contrast, bore no vestiges of sleep history (Extended Data Fig. 6).

AOX protected dFBN mitochondria against sleep loss-induced fragmentation (Fig. 3b and Extended Data Fig. 5e–g), underlining that ROS generation during waking[7] is the initial spark that triggers fission[36,40]. In the same vein, depolarization of dFBNs during mechanical sleep deprivation, which increases ATP consumption by the Na[+]–K[+] pump and thereby reduces the diversion of electrons into ROS (Fig. 2a), preserved the morphology of their mitochondria (Fig. 3b and Extended Data Fig. 5e–g).

Mitochondrial fission is a prelude to the proliferation of mitochondria at contact sites with the endoplasmic reticulum[41] and/or the shedding and clearance of dysfunctional organelle fragments by mitophagy[42] (Fig. 3c). Sleep deprivation stimulated both processes: the reconstitution of fluorescence from GFP fragments anchored in the outer mitochondrial and endoplasmic reticulum membranes (SPLICS_short, whose mere presence had no effect on sleep (Extended Data Fig. 5h)) showed a higher contact site count in dFBNs of sleep-deprived flies (Fig. 3d), whereas mito-QC, a ratiometric sensor detecting the entry

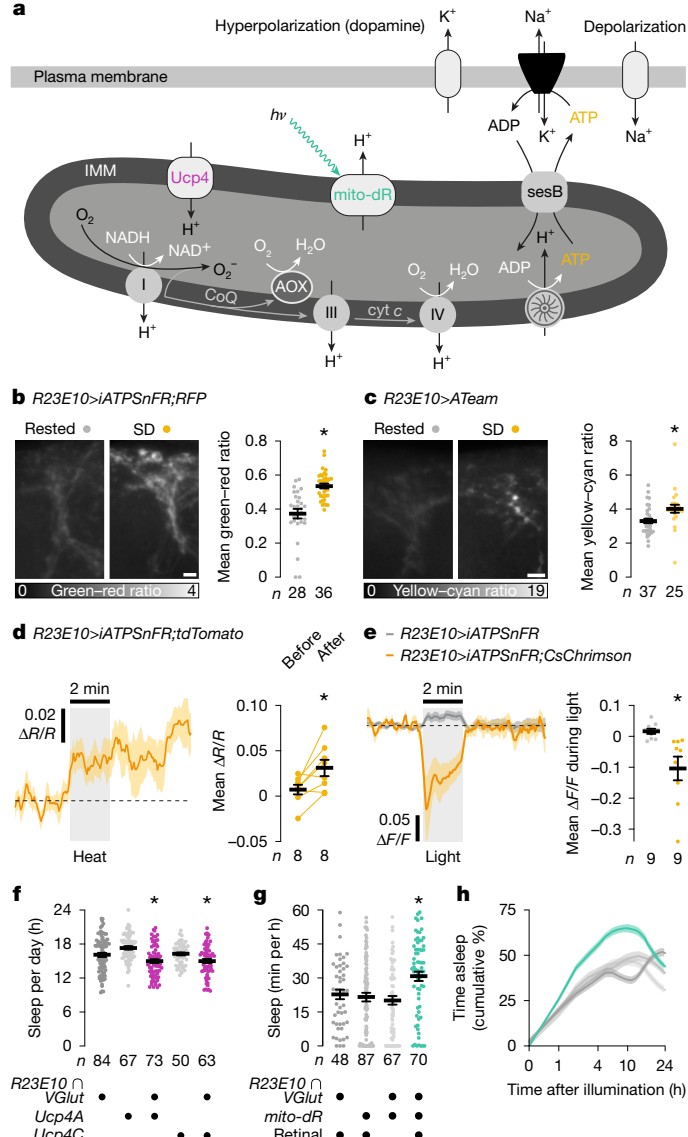

**b** *R23E10>iATPSnFR;RFP*

Rested ● SD ●

0 Green–red ratio 4

Mean green–red ratio

*n* 28 36

**c** *R23E10>ATeam*

Rested ● SD ●

0 Yellow–cyan ratio 19

Mean yellow–cyan ratio

*n* 37 25

**d** *R23E10>iATPSnFR;tdTomato*

2 min

0.02 ΔR/R

Heat

Mean ΔR/R

Before After

*n* 8 8

**e** — *R23E10>iATPSnFR*
— *R23E10>iATPSnFR;CsChrimson*

2 min

0.05 ΔF/F

Light

Mean ΔF/F during light

*n* 9 9

**f**

Sleep per day (h)

*n* 84 67 73 50 63

*R23E10 ∩*
*VGlut*
*Ucp4A*
*Ucp4C*

**g**

Sleep (min per h)

*n* 48 87 67 70

*R23E10 ∩*
*VGlut*
*mito-dR*
*Retinal*

**h**

Time asleep (cumulative %)

Time after illumination (h)

**Fig. 2 | A mitochondrial electron surplus induces sleep. a**, Proton-pumping complexes I, III and IV convert the energy of electron transfers from NADH to $O_2$–through intermediates CoQ and cytochrome *c* (cyt *c*)–into a proton electrochemical gradient, *Δp*, across the IMM. Ucp4 discharges, whereas illumination (*hν*) of mito-dR charges, the IMM. The return of extruded protons to the matrix spins the blades of the ATP synthase and produces ATP, which leaves the matrix via sesB in exchange for cytoplasmic ADP. Neuronal ATP consumption is activity-dependent, in part because the plasma membrane $Na^+$–$K^+$ ATPase must restore ion gradients dissipated by action and excitatory synaptic currents. An oversupply (relative to ATP demand) of electrons to CoQ increases the risk of single-electron reductions of $O_2$ to $O_2^-$ at complexes I and III. AOX mitigates this risk. **b,c**, Summed-intensity projections of dFBN dendrites expressing iATPSnFR plus RFP (**b**) or ATeam (**c**), in rested and sleep-deprived (SD) flies. Emission ratios are intensity-coded according to the keys below and increase after sleep deprivation ($P < 0.0001$ (**b**) and $P = 0.0003$ (**c**); two-sided Mann–Whitney test). **d**, Arousing heat elevates ATP in dFBNs expressing iATPSnFR plus tdTomato. Mean fluorescence was quantified in 20-s windows immediately before and after stimulation ($P = 0.0152$, two-sided paired *t*-test) and is plotted as a change in fluorescence intensity ratio (*ΔR/R*) with co-expressed tdTomato relative to pre-stimulation baseline. **e**, Optogenetic stimulation dissipates ATP in dFBNs expressing iATPSnFR and CsChrimson, but not in dFBNs lacking CsChrimson ($P = 0.0076$, two-sided *t*-test). *ΔF/F* is the change in fluorescence intensity relative to pre-stimulation baseline. **f**, Sleep in flies expressing *R23E10 ∩ VGlut-GAL4*-driven Ucp4A or Ucp4C and parental controls ($P ≤ 0.0381$, Holm–Šídák test after analysis of variance (ANOVA)). **g,h**, Sleep during the first 60 min after illumination (**g**; $P ≤ 0.0432$, Dunn's test after Kruskal–Wallis ANOVA) and cumulative sleep percentages in flies expressing *R23E10 ∩ VGlut-GAL4*-driven mito-dR, with or without retinal, and parental controls (**h**; *Δp* photogeneration effect: $P < 0.0001$, time × *Δp* photogeneration interaction: $P < 0.0001$, mixed-effects model). Asterisks indicate significant differences ($P < 0.05$) from both parental controls or in planned pairwise comparisons. Data are means ± s.e.m.; *n*, number of dendritic regions (**b,c**) or flies (**d**–**h**). Scale bars, 5 μm (**b,c**). For statistical details see Supplementary Table 1.

of mitochondria into acidic autophagolysosomes, reported enhanced mitophagy (Fig. 3e). Mitochondria–endoplasmic reticulum contacts concentrate the fission and fusion machineries[41,43] whose dynamic equilibrium determines the steady-state morphology of the mitochondrial network and support mitochondrial biogenesis by allowing the passage of phospholipids from the endoplasmic reticulum[3,4]. The abundance of mitochondria–endoplasmic reticulum contacts in sleep-deprived dFBNs (Fig. 3d) may thus not only echo a recent wave of mitochondrial fission (Fig. 3a,b and Extended Data Fig. 5c–f), but also prefigure, as we suspect the abundance of transcripts for mitochondrial proteins does (Fig. 1c), the proliferation and fusion of mitochondria during subsequent recovery sleep, which caused their volume, shape and branch length to rebound above baseline values (Fig. 3a,b and Extended Data Fig. 5e, f).

## Mitochondrial dynamics alter sleep

If shifts in the balance between mitochondrial fission and fusion are part of a feedback mechanism that corrects mismatches between NADH supply and ATP demand[36,44,45] that cause sleep pressure to rise or fall, then the experimental induction of these homeostatic responses in dFBNs should move the set points for sleep: mitochondrial fragmentation is predicted to decrease, and mitochondrial fusion is predicted to increase, sleep duration and depth. To test these predictions, we took experimental control of the three GTPases with central regulatory roles

in mitochondrial dynamics (Fig. 4a and Extended Data Fig. 7a–c): the fission dynamin Drp1 (refs. 8–10), and integral proteins of the inner and outer mitochondrial membranes, termed optic atrophy 1 (Opa1) and mitofusin or mitochondrial assembly regulatory factor (Marf), respectively, whose oligomerization in *cis* and *trans* fuses the corresponding membranes[9,11–13]. As in our manipulations of mitochondrial protonmotive force (Fig. 2f–h and Extended Data Fig. 4c–f), *R23E10-GAL4* and the intersectional driver *R23E10 ∩ VGlut-GAL4*, which targets bona fide glutamatergic dFBNs of the central brain[2] (Fig. 1a), were used interchangeably in these experiments (Fig. 4 and Extended Data Fig. 8).

Fragmenting dFBN mitochondria through the overexpression of Drp1 or the RNAi-mediated depletion of Opa1–and, to a lesser extent, of Marf–decreased sleep (Fig. 4b,c and Extended Data Fig. 8a–e), abolished the homeostatic response to sleep deprivation (Fig. 4d and Extended Data Fig. 8f) and reduced ATP concentrations in dFBNs regardless of sleep history (Extended Data Fig. 7d). Tipping the equilibrium towards mitochondrial fusion had the opposite effect: dFBN-restricted knockdown of Drp1 or the overexpression of Opa1 plus Marf–or of Opa1 alone, but not of Marf alone–increased baseline as well as rebound sleep (Fig. 4b–d and Extended Data Fig. 8a,f,g) and elevated the arousal threshold (Extended Data Fig. 9a,b) without causing overexpression artefacts or overt developmental defects (Extended Data Fig. 9c,d). None of these interventions altered sleep when targeted to projection neurons or Kenyon cells (Extended Data Fig. 10). The indiscriminate sleep losses reported[24] after pan-neuronal or glial RNAi knockdowns of either Drp1 or Marf–which in dFBNs alter sleep bidirectionally (Fig. 4b–d and Extended Data Fig. 8a,f), reflecting their antagonistic mitochondria-shaping roles–are difficult to interpret without parallel measurements of mitochondrial (and possibly peroxisomal) form and function. Taken at face value, they could hint that the fusion–fission cycles of many non-dFBN cells are not tied to the sleep–wake cycle but

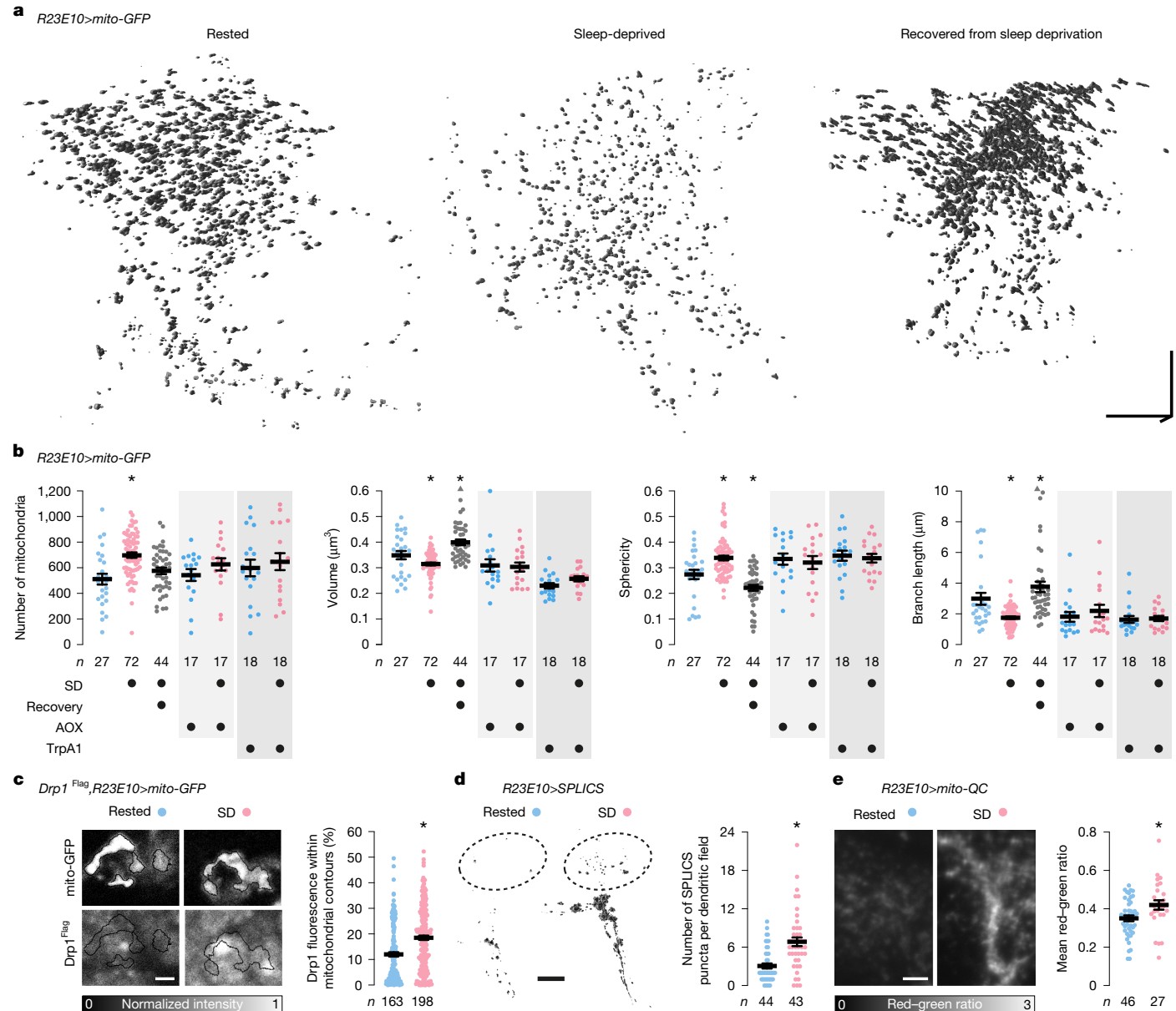

**Fig. 3 | Sleep history alters mitochondrial dynamics. a,b,** Volumetric renderings (**a**) and morphometric parameters (**b**) of automatically detected mitochondria in OPRM image stacks of dFBN dendrites in rested flies, sleep-deprived flies and flies allowed to recover for 24 h after sleep deprivation. Sleep history-dependent changes in mitochondrial number ($P < 0.0001$, Holm–Šídák test after ANOVA), volume ($P = 0.0470$, Dunn's test after Kruskal–Wallis ANOVA), sphericity ($P = 0.0124$, Dunn's test after Kruskal–Wallis ANOVA) and branch length ($P = 0.0033$, Dunn's test after Kruskal–Wallis ANOVA) are occluded by the co-expression of AOX ($P \geq 0.2257$, two-sided $t$-test or Mann–Whitney test) or the simultaneous activation of TrpA1 ($P \geq 0.0625$, two-sided $t$-test or Mann–Whitney test) and (over)corrected after recovery sleep (number of mitochondria: $P = 0.1551$, all other parameters: $P \leq 0.0302$, Dunn's test after Kruskal–Wallis ANOVA). Two data points exceeding the $y$-axis limits are plotted as triangles at the top of the graphs; mean and s.e.m. are based on the actual values. **c,** Drp1 recruitment. Single confocal image planes through dFBN somata of flies expressing *R23E10-GAL4*-driven mito-GFP (top) and Drp1[Flag] from the endogenous locus (bottom). Sleep deprivation increases the percentage of cellular anti-Flag fluorescence (intensity-coded according to the key below) within automatically detected mitochondrial contours ($P < 0.0001$, two-sided Mann–Whitney test). **d,** Mitochondria–endoplasmic reticulum contacts. Isosurface renderings (voxel value 128) of SPLICS puncta in distal dFBN dendritic branches (dashed outlines), obtained by trilinear interpolation of thresholded and despeckled confocal image stacks. Sleep deprivation increases the number of SPLICS puncta per dendritic field ($P < 0.0001$, two-sided Mann–Whitney test). **e,** Mitophagy. Summed-intensity projections of dFBN dendrites expressing mito-QC. Emission ratios are intensity-coded according to the key below and increase after sleep deprivation ($P = 0.0101$, two-sided $t$-test). Data are means ± s.e.m.; $n$, number of dendritic regions (**b,d,e**) or somata (**c**); asterisks, significant differences ($P < 0.05$) in planned pairwise comparisons. Scale bars, 10 μm (**a**), 2 μm (**c**), 10 μm (**d**), 5 μm (**e**). For statistical details see Supplementary Table 1.

operate continuously to mix and re-compartmentalize mitochondrial content for maintenance or metabolic control[44,45].

In dFBNs, the large and opposite behavioural consequences of promoting mitochondrial fission or fusion (Fig. 4b–d) went hand-in-hand with established biophysical signatures of low or high sleep pressure[5,31].

dFBNs in short sleepers overexpressing Drp1 had shallower current–spike frequency functions than neurons in control animals, whereas the converse was true in somnolent overexpressers of Opa1 and Marf (Fig. 4e–g), whose dFBNs generated an elevated number of somnogenic bursts[2] as part of their enhanced responses (Fig. 4e,h).

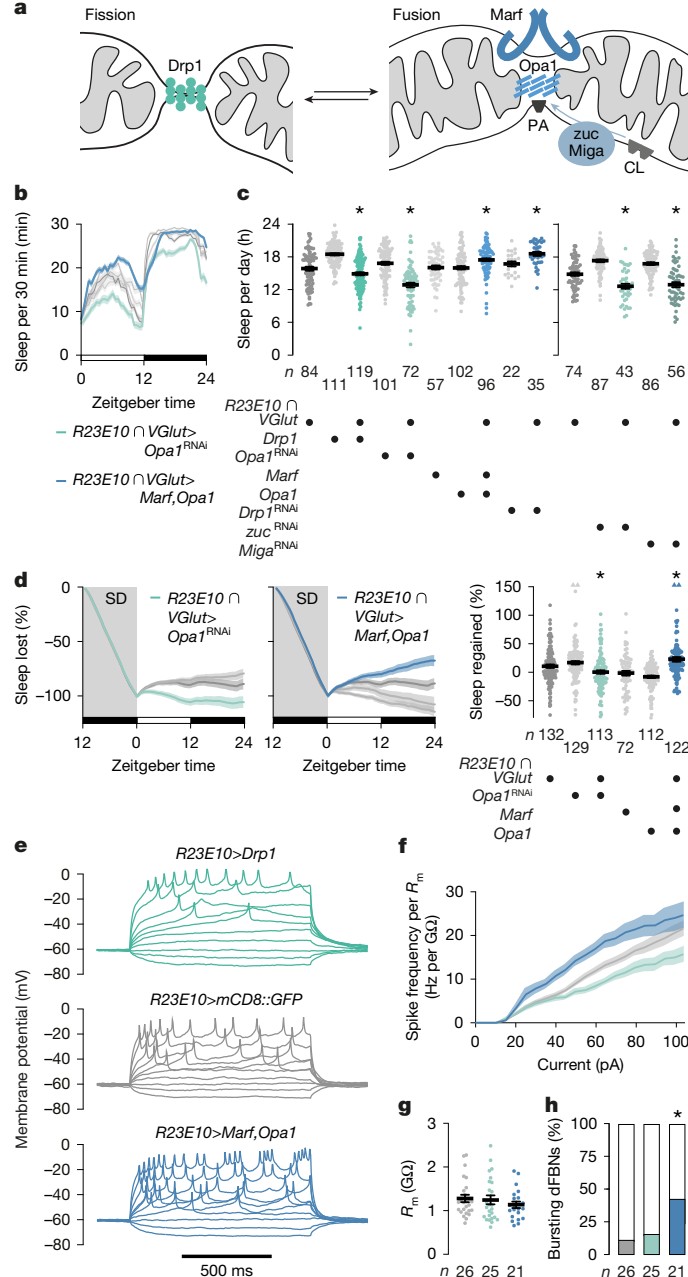

**a** Fission | Fusion Marf

**Fig. 4 | Mitochondrial dynamics alter sleep. a**, The mitochondrial fission (green) and fusion machineries (blue) comprise Drp1, the outer and inner membrane proteins Marf and Opa1, and the mitoPLD zuc, which releases phosphatidic acid (PA) from cardiolipin (CL). Miga stimulates zuc activity and/or supplies phosphatidic acid from other membranes. **b,c**, Sleep profiles (**b**, genotype effect: $P < 0.0001$, time × genotype interaction: $P < 0.0001$, two-way repeated-measures ANOVA) and daily sleep (**c**) in flies expressing $R23E10 \cap VGlut$-GAL4-driven fission or fusion proteins, or RNAi transgenes targeting transcripts encoding these proteins (left) or those regulating phosphatidic acid levels (right), and their parental controls. Manipulations that increase fission (green) or fusion (blue) alter sleep in opposite directions (GTPases: $P \leq 0.0332$, Holm–Šídák test after ANOVA; phosphatidic acid regulators: $P \leq 0.0198$ for all planned pairwise comparisons after Kruskal–Wallis ANOVA). **d**, Manipulations that increase fission ($R23E10 \cap VGlut$-GAL4 > $Opa1^{RNAi}$, green) or fusion ($R23E10 \cap VGlut$-GAL4 > $Marf$,$Opa1$, blue) alter the time courses (left-hand panels, genotype effects: $P \leq 0.0003$, time × genotype interactions: $P < 0.0001$, two-way repeated-measures ANOVA) and percentages of sleep rebound after deprivation in opposite directions (right-hand panel, genotype effect: $P \leq 0.0450$ for all planned pairwise comparisons after Kruskal–Wallis ANOVA). Four data points exceeding the $y$-axis limits are plotted as triangles at the top of the right-hand graph; mean and s.e.m. are based on the actual values. **e**, Example voltage responses to current steps of dFBNs expressing mCD8::GFP (grey) and Drp1 (green) or Marf plus Opa1 (blue). **f**, Manipulations that increase fission (green) or fusion (blue) alter the membrane resistance ($R_m$)-normalized spike frequency in opposite directions (genotype effect: $P < 0.0001$, time × genotype interaction: $P < 0.0001$, mixed-effects model, sample sizes in **g**). **g**, Membrane resistances (genotype effect: $P = 0.4806$, Kruskal–Wallis ANOVA). **h**, The overexpression of Marf plus Opa1 increases the percentage of dFBNs generating bursts of action potentials ($P = 0.0241$, $\chi^2$-test; standardized residuals +2.01). Data are means ± s.e.m.; $n$, number of flies (**b**–**d**) or cells (**f**,**g**). Asterisks, significant differences ($P < 0.05$) in planned pairwise comparisons. For statistical details see Supplementary Table 1.

A striking feature of sleep-deprived brains is the depletion of phosphatidic acid[5], a fusogenic[46] glycerophospholipid. Mitochondrial phosphatidic acid is a cleavage product of cardiolipin, generated by a local phospholipase D (mitoPLD)[46] (Fig. 4a). Underlining the importance of phosphatidic acid for the fusion reaction, and of mitochondrial fusion for the regulation of sleep, $R23E10 \cap VGlut$- or $R23E10$-GAL4-restricted interference with the expression of the mitoPLD zucchini or the outer membrane protein Mitoguardin (Miga), which stabilizes catalytically active mitoPLD[47] and/or transfers phospholipids (including phosphatidic acid) from other cellular membranes to mitochondria[48,49], recapitulated the sleep losses seen when the protein-based fusion machinery of these neurons was targeted by RNAi or antagonized by the overexpression of Drp1 (Fig. 4c and Extended Data Fig. 8b).

## Discussion

Aerobic metabolism was the innovation that, following the first of two large increases in atmospheric $O_2$ levels 2.4 billion and 750–570 million years ago, allowed eukaryotes to maximize the free energy yield of electron transfers, setting the stage for the Cambrian explosion of multicellular life in the wake of the second $O_2$ revolution[50]. Power-hungry nervous systems appeared[51]—and with them, apparently, the need for sleep[52]. Although sleep is likely to have since acquired additional functions, such as synaptic homeostasis or memory consolidation[21], an empirical power law[53] that relates daily sleep amount to mass-specific $O_2$ consumption[54] suggests that sleep serves an ancient metabolic purpose also in mammals. The allometric exponent in this power law is a multiple of $\frac{1}{4}$ rather than the $\frac{1}{3}$ expected from Euclidean geometric scaling—a sign that the distribution of resources by centralized networks, such as the vascular and respiratory systems, is responsible[53,55]. Thanks to higher terminal branch densities, these networks allocate more $O_2$ to each cell in small animals, allowing their metabolism to run 'hotter' than that of large mammals, whose cells are supply-limited[55]. The price to pay is a shorter life, a greater fraction of which is spent asleep. Even within species, variations in the sleep requirements of individuals (including those with identical nuclear genomes, such as our isogenic flies) could arise, in part, from differences in the resistance to electron flow[18] of respiratory complexes containing subunits encoded by mitochondrial DNA. An overwhelming sense of tiredness (which is unrelated to muscle fatigue) is in fact a common symptom of human mitochondrial disease[56].

If sleep indeed evolved to fulfil a metabolic need, it is not surprising that neurons controlling sleep and energy balance would be regulated by similar mechanisms. In the mammalian hypothalamus, the mitochondria of orexigenic neurons expressing agouti-related protein (AgRP) and of anorexigenic neurons expressing pro-opiomelanocortin undergo antiphasic cycles of fission and fusion[57]. These cycles are coupled to changes in the energy balance of mice[57], just as cycles of mitochondrial fission and fusion in dFBNs are coupled to changes in the sleep balance of flies. The electrical output of AgRP neurons increases after mitochondrial fusion to promote weight gain and fat deposition[57],

just as the electrical output of dFBNs increases after mitochondrial fusion to promote sleep. Deletions of mitofusins from AgRP neurons impair the consumption of food[57], just as interference with mitochondrial fusion in dFBNs impairs the induction of sleep. These parallels suggest that sleep pressure and hunger both have mitochondrial origins, and that electrons flow through the respiratory chains of the respective feedback controllers like sand in the hourglass that determines when balance must be restored.

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

## Methods

### *Drosophila* strains and culture

Flies were grown on media of cornmeal (62.5 g l⁻¹), inactive yeast powder (25 g l⁻¹), agar (6.75 g l⁻¹), molasses (37.5 ml l⁻¹), propionic acid (4.2 ml l⁻¹), tegosept (1.4 g l⁻¹) and ethanol (7 ml l⁻¹) under a 12 h light–12 h dark cycle at 25 °C in approximately 60% relative humidity, unless stated otherwise. To prevent the undesired activation of optogenetic actuators or the photoconversion of all-*trans* retinal by ambient light, flies expressing CsChrimson or mito-dR and their controls were reared and housed in constant darkness and transferred to food supplemented with 2 mM all-*trans* retinal (Molekula) in dimethylsulfoxide (DMSO), or to DMSO vehicle only, 2 days before the optical stimulation experiments, at an age of 1–2 days post eclosion. Flies expressing TrpA1 and their controls were cultured and maintained at 21 °C and shifted to 29 °C for 12 h.

Transgene expression was directed to dFBNs by driver lines *R23E10-GAL4* (ref. 58) or *R23E10 ∩ VGlut-GAL4* (ref. 2) (created by reconstituting GAL4 from hemidrivers[59] *R23E10-DBD*[60] and *VGlut-p65AD*[61]); projection neurons, Kenyon cells and dopaminergic neurons were targeted by *GH146-GAL4* (ref. 62), *OK107-GAL4* (ref. 63) and *TH-LexA*[64], respectively. Effector transgenes encoded fluorescent markers for flow cytometry (*UAS-6xEGFP*[65]), visually guided patch-clamp recordings (*UAS-mCD8::GFP*[66]), mitochondrial morphometry (*UAS-mito-GFP*[67,68]) or ratiometric imaging (*UAS-RFP* or *UAS-tdTomato*); integral proteins of the outer mitochondrial membrane[69] (*UAS-OMM-mCherry*), endoplasmic reticulum[70] (*UAS-tdTomato-Sec61β*) or plasma membrane[71] (*UAS-CD4-tdTomato*); the ATP sensors iATPSnFR[1.0] (refs. 72,73) or ATeam1.03NL[74,75]; the mitochondrial alternative oxidase AOX of *Ciona intestinalis*[76]; delta-rhodopsin of *Haloterrigena turkmenica* with an IMM-targeting sequence[16,77] (mito-dR); the opto- or thermogenetic actuators[78] CsChrimson[79] or TrpA1 (the latter in *UAS-* and *lexAop*-driven versions[80,81]); the mitochondria–endoplasmic reticulum contact site[82,83] or mitophagy[84,85] reporters SPLICS[short] or mito-QC; overexpression constructs encoding Ucp4A or Ucp4C[86,87], Drp1 (3 independent transgenes[88–90]), Opa1 (ref. 89) or Marf (2 independent transgenes[89,91]); or RNAi transgenes for interference with the expression of *Drp1* (ref. 91), *Opa1* (7 independent transgenes[88,92,93]), *Marf* (5 independent transgenes[88,92,93]), *zucchini*[93] or *Mitoguardin* (2 independent transgenes[92]). Recombinant strains carrying the *UAS-Marf* and *R23E10-GAL4* or *R23E10-DBD* transgenes on the third chromosome were generated to enable the co-expression of Opa1 and Marf. Endogenous Drp1 was colocalized with mitochondria in *Drp1::FLAG-FlAsH-HA* flies[94]. Supplementary Table 3 lists all fly strains and their sources.

### Sleep measurements and sleep deprivation

In standard sleep assays, females aged 2–4 days were individually inserted into 65-mm glass tubes containing food reservoirs, loaded into the Trikinetics *Drosophila* Activity Monitor system and housed under 12 h light–12 h dark conditions at 25 °C in 60% relative humidity. Flies were allowed to habituate for 1 day before sleep—classified as periods of inactivity lasting more than 5 min[95,96] (Sleep and Circadian Analysis MATLAB Program[97])—was averaged over two consecutive recording days. Immobile flies (fewer than two beam breaks per 24 h) were manually excluded.

Our standard method of sleep deprivation used the sleep-nullifying apparatus[98]: a spring-loaded platform stacked with Trikinetics monitors was slowly tilted by an electric motor, released and allowed to snap back to its original position. The mechanical cycles lasted 10 s and were repeated continuously for 12 h, beginning at zeitgeber time 12. Flies expressing TrpA1 in dFBNs (and relevant controls) were mechanically sleep-deprived at 29 °C.

To guard against potential side effects of regular mechanical agitation, we explored two alternative methods of sleep deprivation, with comparable results. In some behavioural experiments (Extended Data Fig. 8f), an Ohaus Vortex Mixer stacked with Trikinetics monitors produced horizontal circular motion stimuli with a radius of about 1 cm at 25 Hz for 2 s; in contrast to the rhythmic displacement of flies in the sleep-nullifying apparatus[98], stimulation periods were randomly normally distributed in 20-s bins, hindering adaptation. In some morphometric experiments, flies expressed TrpA1 in arousing dopaminergic neurons[14], whose activation at 29 °C produced sleep deprivation without sensory stimulation (Extended Data Fig. 5c,d).

Rebound sleep was measured in the 24-h window after deprivation. Cumulative sleep loss was calculated for each individual by comparing the percentage of sleep lost during overnight sleep deprivation with the immediately preceding unperturbed night. Individual sleep regained was quantified by normalizing the difference in sleep amount between the rebound and baseline days to baseline sleep. Only flies losing more than 95% of baseline sleep were included in the analysis.

Arousal thresholds were determined by applying horizontal circular motion stimuli with a radius of -1 cm at 8 Hz, generated by a Talboys Multi-Tube Vortexer. Stimuli lasting 0.5–20 s were delivered once every hour between zeitgeber times 0 and 24, and the percentages of sleeping flies (if any) awakened within 1 min of each stimulation episode were quantified.

### Sleep induction by photo-energized mitochondria

For experiments with photo-energized mitochondria[16,77], 3–5-day-old females expressing *R23E10 ∩ VGlut-GAL4-* or *R23E10-GAL4*-driven mito-dR and their parental controls were reared for 2 days on standard food supplemented with all-*trans* retinal (or DMSO vehicle only, as indicated) and individually transferred to the wells of a flat-bottom 96-well plate. Each well contained 150 μl of sucrose food (5% sucrose, 1% agar) with or without 2 mM all-*trans* retinal. The plate was sealed with a perforated transparent lid, inserted into a Zantiks MWP Z2S unit operated at 25 °C and illuminated from below by infrared LEDs while a camera captured 31.25 frames per s from above. Zantiks software extracted time series of individual movements, which were converted to sleep measurements with the help of a custom MATLAB script detecting continuous stretches of zero-speed bins lasting more than 5 min. After flies had been allowed to habituate for at least one day in the absence of stimulation light, high-power LEDs running on an 80% duty cycle at 2 Hz (PWM ZK-PP2K) delivered about 7 mW cm⁻² of 530-nm light for 1 h. Movement was monitored for 24 h, including the initial hour of optogenetic stimulation. If flies were found dead at the end of the experiment, data from the 1-h time bin preceding the onset of continuous immobility (>98% zero-speed bins during two or more consecutive hours until the end of the recording) onward were excluded. The >98% threshold was applied to avoid scoring rare instances of video tracker noise as movement.

### Brain dissociation and cell collection for scRNA-seq

On each experimental day, averages of 186 rested and 144 sleep-deprived female flies were retrieved alive from Trikinetics monitors (Extended Data Fig. 1a) and dissected in parallel in ice-cold Ca²⁺- and Mg²⁺-free Dulbecco's PBS (Thermo Fisher) supplemented with 50 μM D(-)-2-amino-5-phosphonovalerate, 20 μM 6,7-dinitroquinoxaline-2-3-dione and 100 nM tetrodotoxin (tDPBS) to block excitatory glutamate receptors and voltage-gated sodium channels. The lower number of sleep-deprived flies recovered reflects accidental mortality associated with the operation of the sleep-nullifying apparatus. Brains were transferred to Protein LoBind microcentrifuge tubes containing ice-cold Schneider's medium supplemented with the same toxins (tSM), washed once with 1 ml of tSM, incubated in tSM containing 1.11 mg ml⁻¹ papain and 1.11 mg ml⁻¹ collagenase I for 30 min at room temperature, washed again with tSM and subsequently triturated with flame-rounded 200-μl pipette tips. Dissociated brain cells were collected by low-speed centrifugation (2,000 rpm, 3 min), resuspended in 1 ml tDPBS, and filtered through a 20-μm CellTrics strainer. For the isolation of cells by flow cytometry, dead cells were excluded with the help of a DAPI viability

dye (1 µg ml$^{-1}$; BD Pharmingen). Single cells were gated for on the basis of forward and side scatter parameters, followed by subsequent gating for EGFP-positive and EGFP-negative cells, using the integrated fluorescence excited at 488 nm and collected in the 500–526-nm band (Becton Dickinson FACSDiva software). Both the EGFP-positive fraction and the EGFP-negative flow-through were collected for sequencing. Samples were partitioned into single cells and barcoded using droplet microfluidics[30] (10X Chromium v.3 and v.3.1) and multiplexed during Illumina NovaSeq6000 sequencing. Brains and dissociated cells were kept on ice or at ice-cold temperatures from dissection to sample submission, including during flow cytometry, but not during the enzymatic and mechanical dissociation steps, which took place at room temperature.

## scRNA-seq data processing and alignment
Raw transcriptomic data were pre-processed with a custom command line script[30,99,100], which extracted cell barcodes and aligned associated reads to a combination of the *Drosophila melanogaster* genome release BDGP6.22 and the reference sequences of the *GAL4* and *EGFP-p10$^{3'UTR}$* transgenes, using STAR 2.6.1b with default settings[101]. Flybase version FB2018_03 gene names were used for annotation. The cumulative fraction of reads as a function of cell barcodes, arranged in descending order of the number of reads, was inspected, and only cells with a high number of reads, up to a clearly visible shoulder, were retained[30]; beads with few reads, potentially ambient RNA, were discarded. All subsequent analyses were performed in R, using the Seurat v.4.1 package[102].

Three biological replicates, collected on different days from independent genetic crosses, were merged, variation driven by individual batches was regressed out, and the data were normalized by dividing by the number of unique molecular identifiers per cell and multiplying by 10,000. Applying standard criteria for fly neurons[28,100,103,104], we rejected genes detected in fewer than three cells and retained only cells associated with 800–10,000 unique molecular identifiers and 200–5,000 transcripts.

Principal component analysis was used to compress the expression data from an initial dimensionality of 10,000 (the number of variable features) to 50 (the number of principal components we chose to consider); the scores along these 50 dimensions were then visualized in a two-dimensional uniform manifold approximation and projection embedding. Clusters were identified by constructing a shared nearest neighbour graph and applying the Louvain algorithm with resolution 0.2. Clusters were annotated manually according to the presence of established markers[100,103]: cholinergic, glutamatergic and GABAergic neurons expressed *elav* and *nSyb* and, respectively, genes encoding VAChT or the glutamate transporter (VGlut) or glutamic acid decarboxylase 1 (Gad1) at levels >2; Kenyon cells were identified by the expression of *eyeless* and *Dop1R2* and partitioned into αβ, α′β′ and γ divisions according to the distributions of *sNPF*, *fasciclin 2* and *trio*; monoaminergic neurons were identified by the presence of the vesicular monoamine transporter Vmat and divided into dopaminergic, serotonergic and octopaminergic–tyraminergic neurons by the co-expression of genes encoding biosynthetic enzymes (dopa decarboxylase, tyrosine hydroxylase, tyrosine decarboxylase 2, tryptophan hydroxylase and tyramine β-hydroxylase) and vesicular transporters for dopamine or serotonin (DAT and SerT); projection neurons were defined and classified by the expression of the transcription factors cut and abnormal chemosensory jump 6 (acj6), with or without Lim1; glia lacked elav and nSyb but expressed the Na$^+$–K$^+$ ATPase encoded by the *nervana 2* gene, whereas astrocytes also contained the astrocytic leucine-rich repeat molecule (alrm); cells of the fat body were recognized by the expression of *Secreted protein, acidic, cysteine-rich* (*SPARC*), *Metallothionein A* (*MtnA*), *I'm not dead yet* (*Indy*) and *pudgy*; *R23E10* neurons expressed *elav* and *nSyb* plus *AstAR-1* and *EGFP* at a level >2. Expression level cutoffs for *VAChT*, *VGlut*, *Gad1* and *EGFP* were chosen to bisect bimodal distributions (Extended Data Fig. 1e). For reclustering neurons expressing specific fast-acting neurotransmitters, dFBNs, Kenyon cells or projection neurons, 150 genes were used as variable features.

## Differential gene expression and gene ontology analyses
Differentially expressed genes were identified in Seurat by means of the 'FindMarkers' function, using the 'RNA assay' counts of the two comparison groups, and restricted to genes detected in ≥1% of cells in either of the two groups with, on average, a ≥0.01-fold (log scale) expression level difference between the rested and sleep-deprived states. Expression levels were compared by Bonferroni-corrected Wilcoxon rank-sum test.

Gene ontology terms enriched in the set of differentially expressed nuclear genes were identified using PANTHER v.17 or the ViSEAGO v.1.4.0 and topGO v.2.42.0 packages. PANTHER compared the list of differentially expressed genes with the *Drosophila melanogaster* reference list and the gene ontology annotation database (https://doi.org/10.5281/zenodo.7942786, version 2023-05-10). In ViSEAGO and topGO, differentially expressed genes were compared to a reference set of all variable genes used in Seurat and annotated in Ensembl; only gene ontology terms with more than 40 attached genes were considered, and enriched terms with unadjusted $P < 0.001$ were clustered hierarchically according to Wang's distance[105].

## Two-photon imaging
Females aged 3–4 days were head-fixed to a custom mount with eicosane (Sigma) and imaged on Movable Objective Microscopes with galvanometric or resonant scanners (Sutter Instruments) controlled through ScanImage v.5.4.0 software (Vidrio Technologies). Cuticle, adipose tissue and trachea were removed to create an optical window, and the brain was superfused with carbogenated extracellular solution (95% $O_2$–5% $CO_2$, pH 7.3, 275 mOsm) containing 103 mM NaCl, 3 mM KCl, 5 mM TES, 8 mM trehalose, 10 mM glucose, 7 mM sucrose, 26 mM NaHCO$_3$, 1 mM NaH$_2$PO$_4$, 1.5 mM CaCl$_2$ and 4 mM MgCl$_2$. To excite iATPSnFR and co-expressed tdTomato, Mai Tai DeepSee (Spectra Physics model eHP DS) or Coherent Chameleon Ultra II Ti:sapphire lasers produced 930-nm excitation light pulses whose power was modulated by Pockels cells (302RM, Conoptics). Emitted photons were collected by ×20 (1.0 numerical aperture (NA)) water immersion objectives (W-Plan-Apochromat, Zeiss), split into green and red channels by dichromatic mirrors (Chroma 565dcxr or Semrock BrightLine 565 nm) and detected by GaAsP photomultiplier tubes (H10770PA-40 SEL, Hamamatsu Photonics). The emission paths contained bandpass filters for iATPSnFR (Semrock BrightLine FF01-520/60) and tdTomato (Chroma ET605/70 m), respectively. Photocurrents were passed through high-speed amplifiers (HCA-4M-500K-C, Laser Components) and integrators (BLP-21.4+, Mini-Circuits, or EF506, Thorlabs) to maximize the signal-to-noise ratio.

To gate open the conductance of CsChrimson, a 625-nm LED (M625L3, ThorLabs) controlled by a dimmable LED driver (ThorLabs) delivered 0.5–25 mW cm$^{-2}$ of optical power through a bandpass filter (Semrock BrightLine FF01-647/57-25) to the head of the fly. Stimulus trains lasted for 2 min and consisted of ten 25-ms light pulses in 500-ms bursts recurring once per s. To apply arousing heat, an 808-nm laser diode (Thorlabs L808P500MM) was mounted on a temperature-controlled heat sink (ThorLabs TCDLM9 with ThorLabs TED200C controller) and aimed at the abdomen of the fly. The diode was restricted to a maximal output of 50 mW by a ThorLabs LDC210C laser diode controller, and 2-s pulses were delivered every 30 s for 2 min. Images of 256 × 256 pixels were acquired at a rate of 29.13 Hz. The voltage steps controlling the LED or laser diode were recorded in a separate imaging channel for post-hoc alignment.

To drive proton pumping by mito-dR, a 530-nm LED (M530L3, ThorLabs) controlled by a dimmable LED driver (ThorLabs) delivered about 25 mW cm$^{-2}$ of optical power through an ACP2520-A collimating lens (ThorLabs) to the head of the fly. Stimuli lasted for 400 ms

and were repeated every 10 s. Two shutters, one in the combined fluorescence emission path (∅1/2″ stainless steel diaphragm optical beam shutter with controller, Thorlabs) and one on the LED (Vincent/UniBlitz VS35S2ZM1R1-21 Uni-Stable Shutter with UniBlitz VMM-T1 Shutter Driver/Timer controller), opened alternately during imaging and green light stimulation. Images of 128 × 50 pixels were acquired at a rate of 41.59 Hz. The voltage steps controlling the LED were recorded in a separate imaging channel for post-hoc alignment.

Time series of average fluorescence in manually selected dendritic regions of interest were analysed in MATLAB, following the subtraction of a time-varying background. $\Delta F/F$ curves were calculated separately for each trial as $\frac{\Delta F_t}{F_0} = \frac{(F_t - F_0)}{F_0}$, where $F_0$ is the mean fluorescence intensity before stimulation onset (170 s for heat and CsChrimson, 4 s for mito-dR) and $F_t$ is the fluorescence intensity in frame $t$; $\frac{\Delta R_t}{R_0}$ represents the iATPSnFR–tdTomato (green–red) intensity ratio. The stimulus-aligned $\frac{\Delta R_t}{R_0}$ signals were averaged across two trials in the case of CsChrimson and 30 trials in the case of mito-dR and then across flies; the statistical units are flies. For display purposes, traces were smoothed with moving-average filters (15-s windows for heat and CsChrimson, followed by downsampling by a factor of 100; 1-s window for mito-dR).

## Super-resolution and confocal imaging
Single-housed females aged 6 days post eclosion were dissected at zeitgeber time 0, following ad libitum sleep or 12 h of sleep deprivation. Experimental and control samples were processed in parallel. Brains were fixed for 20 min in 0.3% (v/v) Triton X-100 in PBS (PBST) with 4% (w/v) paraformaldehyde, washed five times with PBST, incubated with primary antibodies and secondary detection reagents where indicated, mounted in Vectashield and imaged. Only anatomically intact specimens from live flies (at the point of dissection) were analysed, blind to sleep history, using existing, adapted or newly developed (semi-)automated routines in Fiji 2.14.0/1.54f. The specific acquisition and analysis parameters for different experiments are as follows and further detailed in Supplementary Table 4.

For mitochondrial morphometry[106], $z$-stacks of the dendritic fields of mito-GFP-expressing dFBNs (or the glomerular arborizations of projection neurons) were collected at the Nyquist limit, with identical image acquisition settings across all conditions. Dendrites were chosen as an optically favourable compartment for analysis because their branches (and the mitochondria within them) were well separated along all axes.

OPRM super-resolution images were collected on an Olympus IX83 P2ZF microscope (Olympus cellSense Dimension 4.3.1) equipped with a UplanSApo ×60 (1.30 NA) silicon oil objective and a Yokogawa CSU-W1 super-resolution by optical pixel reassignment spinning-disc module[38]. Image stacks were passed through a 'low' Olympus Super Resolution spatial frequency filter and deconvolved in five iterations of a constrained maximum likelihood algorithm (Olympus cellSense Dimension 4.3.1).

CLSM images were acquired on a Leica TCS SP5 confocal microscope (Leica LAS AF 2.7.3.9723) with an HCX IRAPO L ×25 (0.95 NA) water immersion objective. Point-spread functions were created with the PSF Generator plugin[107] in Fiji and used to deconvolve the images with DeconvolutionLab2 2.1.2 software[108,109] using the Richardson–Lucy algorithm with total variation regularization (set to 0.0001) and a maximum of two iterations[106].

Functions of the Mitochondria Analyzer 2.3.1 plugin in Fiji were applied in an automated fashion to the deconvolved OPRM and CLSM images to remove background noise ('subtract background' with a radius of 1.25 µm), reduce noise and smooth objects while preserving edges ('sigma filter plus'), enhance dim areas while minimizing noise amplification ('enhance local contrast' with slope 1.4) and optimize the use of image bits ('gamma correction' with a value of 0.90) before thresholding ('weighted mean' with block size 1.25 µm and a 'C value' of 5 for dFBNs and 12 for projection neurons, determined empirically to minimize background noise[106]). The resulting binary images were

examined by means of Batch 3D analysis on a 'per-cell' basis to extract morphological metrics; OPRM image stacks were rendered in three dimensions with the Volume Viewer 2.02 plugin in Fiji, using trilinear interpolation in volume mode with default settings and a greyscale transfer function. The full dendritic fields of dFBNs were analysed, but the high packing density of projection neuron dendrites in the glomeruli of the antennal lobe forced us to select 20 substacks per glomerulus, each with an axial depth of 6 µm (for OPRM) or 5.8 µm (for CLSM), using a Fiji random number generator function, to make computations practical. Morphometric parameters were then averaged across all substacks per glomerulus.

When other *UAS*-driven transgenes were present in addition to *UAS-mito-GFP*, the dilution of a limiting amount of GAL4 among several promoters led to a noticeable dimming of mitochondrial fluorescence, especially in OPRM images. We therefore sought to compare mitochondrial morphologies within (Fig. 3a,b and Extended Data Figs. 5g and 6a,b) or among (Extended Data Figs. 5c,d and 7a,b) genotypes carrying the same number of *UAS*-driven transgenes. Although shape metrics, such as the mean volume, sphericity and branch length of mitochondria, seemed robust under variations in brightness, the number of fluorescent objects crossing the detection threshold was not. Because the absolute number of mitochondria in an image stack also depends on the volume fraction occupied by dFBN dendrites, these limitations should be borne in mind when interpreting mitochondrial counts.

For ratiometric snapshot imaging of the ATP sensors iATPSnFR (normalized to co-expressed RFP) and ATeam and the mitophagy sensor mito-QC, fluorescence was quantified on summed $z$-stacks of the relevant channels, following the subtraction of average background in manually defined areas close to the structures of interest. Brains expressing ATeam were imaged on a Zeiss LSM980 with Airyscan2 microscope (ZEN blue 3.3) with a Plan-Apochromat ×40 (1.30 NA) oil immersion objective, an excitation wavelength of 445 nm for the FRET donor mCFP, and emission bands of 454–507 nm for CFP and 516–693 nm for the YFP variant mVenus[75]. In ATP calibration experiments (see below), the biocytin-filled dFBN soma was identified after staining with Alexa Fluor 633 streptavidin (ThermoFisher, 1:600 in 0.3% PBST at 4 °C overnight) at excitation and emission wavelengths of 639 nm and 649–693 nm, respectively. Images of iATPSnFR plus RFP and mito-QC were acquired on a Leica TCS SP5 confocal microscope with HCX IRAPO L ×25 (0.95 NA) water and HCX PL APO ×40 (1.30 NA) oil immersion objectives, respectively. The excitation and emission wavelengths were 488 nm and 498–544 nm for iATPSnFR, 555 nm and 565–663 nm for RFP, and 488 nm and 503–538 nm, and 587 nm and 605–650 nm, for the GFP and mCherry moieties of mito-QC, respectively.

To convert the emission ratios of ATeam1.03NL into approximate ATP concentrations, we constructed a calibration curve in vivo by including 0.15, 1.5 or 4 mM MgATP, along with 10 mM biocytin, in the intracellular solution during whole-cell patch-clamp recordings. After dialysing the recorded dFBN for 30 min, the patch electrode was withdrawn, and the brain was recovered, fixed and processed for detection and ratiometric imaging of the biocytin-labelled soma. A standard curve was obtained by fitting the Hill equation, with a dissociation constant[75] of 1.75 mM and a Hill coefficient[74] of 2.1, to the YFP–CFP fluorescence ratios of dFBN somata containing known ATP concentrations.

SPLICS puncta[82,110] were imaged on a Leica TCS SP5 confocal microscope with an HCX PL APO ×40 (1.30 NA) oil immersion objective, excitation and emission wavelengths of 488 and 500–540 nm, respectively, and analysed in Fiji with a custom macro based on the Quantification 1 and 2 plugins[110]. Only puncta exceeding 10 voxels were counted.

For localizing Drp1, we labelled the mitochondria of dFBNs with mito-GFP in *Drp1::FLAG-FlAsH-HA* flies[94], whose genomic *Drp1* coding sequence is fused in frame with a Flag-FlAsH-HA tag. Fixed brains were incubated sequentially at 4 °C in blocking solution (10% goat serum in 0.3% PBST) overnight, with mouse monoclonal anti-Flag antibody (anti-DDK; 1:1,000, OriGene) in blocking solution for 2–3 days, and with

goat anti-Mouse Alexa Fluor 633 (1:500, ThermoFisher) in blocking solution for two days. The samples were washed five times with PBST before and after the addition of secondary antibodies, mounted and imaged on a Leica TCS SP5 confocal microscope with an HCX PL APO ×40 (1.30 NA) oil immersion objective. The excitation and emission wavelengths were 488 and 500–540 nm for mito-GFP and 631 and 642–690 nm for Alexa Fluor 633, respectively. dFBN somata were identified manually in the green channel, which was then thresholded, despeckled and binarized by an automated custom macro in Fiji. The mitochondria-associated fraction of endogenous Flag-tagged Drp1 in dFBNs was quantified in summed $z$-stacks as the proportion of red-fluorescent pixels in the somatic volume that colocalized with mitochondrial objects.

### Electrophysiology

For whole-cell patch-clamp recordings in vivo, female flies aged 2–4 days post eclosion were prepared as for functional imaging, but the perineural sheath was also removed for electrode access. The GFP-labelled somata of dFBNs were visually targeted with borosilicate glass electrodes (8–10 MΩ) filled with internal solution (pH 7.3, 265 mOsm) containing: 10 mM HEPES, 140 mM potassium aspartate, 1 mM KCl, 4 mM MgATP, 0.5 mM Na₃GTP, 1 mM EGTA and 10 mM biocytin. Signals were acquired at room temperature (23 °C) in current-clamp mode with a MultiClamp 700B amplifier (Molecular Devices), lowpass-filtered at 5 kHz, and sampled at 10 kHz using an Axon Digidata 1550B digitizer controlled through pCLAMP 11.2 (Molecular Devices). Series resistances were monitored but not compensated. Data were analysed using v.3.0c of the NeuroMatic package[111] (http://neuromatic.thinkrandom.com) in Igor Pro 8.04 (WaveMetrics). Current–spike frequency functions were determined from voltage responses to a series of current steps (5-pA increments from −20 to 105 pA, 1 s duration) from a pre-pulse potential of −60 ± 5 mV. Spikes were detected by finding minima in the second derivative of the membrane potential trace. Spike frequencies were normalized to membrane resistances, which were calculated from linear fits of the steady-state voltage changes elicited by hyperpolarizing current steps. Only dFBNs firing more than one action potential in response to depolarizing current injections, with resting potentials below −30 mV and series resistances remaining below 50 MΩ throughout the recording, were characterized further. Spike bursts were defined as sets of spikes with an average intra-burst inter-spike interval (ISI) less than 50 ms and an inter-burst ISI more than 100 ms. These ISI thresholds were set after visual inspection of the voltage traces of all recorded neurons. Cells were scored as bursting if they generated at least one action potential burst during the series of depolarizing current steps.

### Analysis of mitochondria in volume electron micrographs

The hemibrain v.1.2.1 connectome[39] was accessed in neuPrint+ (ref. 112). The volumes of mitochondria in all $R23E10\text{-}GAL4$-labelled dFBNs[113] and uniglomerular olfactory projection neurons[39] in the left hemisphere (which contains mostly the dendritic compartments of projection neurons) were retrieved through Neo4j Cypher and neuPrint-python queries and analysed with custom MATLAB scripts. Supplementary Table 5 lists the identification numbers (Body_IDs) of all analysed cells.

### Quantification and statistical analysis

With the exception of sleep measurements, no statistical methods were used to predetermine sample sizes. Flies of the indicated genotype, sex and age were selected randomly for analysis and assigned randomly to treatment groups if treatments were applied (for example, sleep deprivation). The investigators were blind to sleep history and/or genotype in imaging experiments but not otherwise. All behavioural experiments were run at least three times, on different days and with different batches of flies. The figures show pooled data from all replicates.

Gene expression levels were compared by Bonferroni-corrected two-sided Wilcoxon rank-sum test. Gene ontology enrichment was quantified using Fisher's exact test in PANTHER (with a false discovery rate-adjusted significance level of $P < 0.05$) or ViSEAGO (with an unadjusted significance level of $P < 0.001$).

Behavioural, imaging and electrophysiological data were analysed in Prism v.10 (GraphPad) and SPSS Statistics 29 (IBM). All null hypothesis tests were two-sided. To control type I errors, $P$ values were adjusted to achieve a joint $\alpha$ of 0.05 at each level in a hypothesis hierarchy; multiplicity-adjusted $P$ values are reported in cases of several comparisons at one level. Group means were compared by $t$-test, one-way ANOVA, two-way repeated-measures ANOVA or mixed-effects models, as stated, followed by planned pairwise analyses with Holm–Šídák's multiple comparisons test where indicated. Repeated-measures ANOVA and mixed-effect models used the Geisser–Greenhouse correction. Where the assumption of normality was violated (as indicated by D'Agostino–Pearson test), group means were compared by Mann–Whitney test or Kruskal–Wallis ANOVA, followed by planned pairwise analyses using Dunn's multiple comparisons test, as indicated. Frequency distributions were analysed by $\chi^2$-test, and categories responsible for pairwise differences[114] were detected by locating cells with standardized residuals ≥2.

### Reporting summary

Further information on research design is available in the Nature Portfolio Reporting Summary linked to this article.

## Data availability

Single-cell transcriptomic data can be found in NCBI's Gene Expression Omnibus repository under accession number GSE256379. Source data are provided with this paper.

## Code availability

The code used to analyse transcriptome data is available in the Drop-seq Alignment Cookbook via GitHub at https://github.com/broadinstitute/Drop-seq and at https://satijalab.org/seurat/.

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

**Acknowledgements** We thank C. Talbot for instrumentation; L. Ballenberger, K. Christodoulou, C. Hartmann, A. Krebbers, Y. Siu and T. Wong for help with genetics and dissections; W. Budenberg for support in customizing the Zantiks MWP unit; the Don Mason Facility of Flow Cytometry and the Light Microscopy Facility, Sir William Dunn School of Pathology, University of Oxford, for cell sorting and access to the Olympus SoRa microscope, respectively; the Oxford Genomics Centre at the Wellcome Centre for Human Genetics (funded by Wellcome grant 203141/A/16/Z) for RNA sequencing; the Oxford Micron Bioimaging Facility (funded by Wellcome grants 091911/B/10/Z and 107457/Z/15/Z) for access to the Zeiss LSM980 microscope; H. Bellen, S. Bullock, J. Chung, T. Clandinin, R. Davis, M. Feany, M. Guo, Y. Imai, H. Jacobs, Y. Jan, V. Jayaraman, L. Luo, J. McNew, G. Rubin, W. Saxton, R. Stowers, S. Waddell, A. Whitworth, the Bloomington Stock Center, the Vienna *Drosophila* Resource Center, the Transgenic RNAi Project (TRiP) and the FlyORF Zurich ORFeome Project for flies; V. Croset, C. Treiber and S. Waddell for advice on single-cell RNA sequencing; the FlyLight Project Team at Janelia Research Campus for anatomical images reproduced in Extended Data Fig. 1h; and T. Cali, P. Hasenhuetl, R. Klemm, V. Savage and E. Vrontou for discussions. This work was supported by grants from the European Research Council (832467) and Wellcome (209235/Z/17/Z and 106988/Z/15/Z) to G.M.; R.S. held a Wellcome Four-Year PhD Studentship in Basic Science (215200/Z/19/Z); AK. received postdoctoral fellowships from the Swiss National Science Foundation and EMBO.

**Author contributions** R.S. performed and analysed all transcriptomic, imaging and behavioural experiments. C.D.V. performed and analysed all electrophysiological recordings. R.S. analysed volume electron microscopy data. N.M. assisted with imaging, genetics and behaviour. A.K. conducted initial studies with Ucp4. R.S. and G.M. designed the study, interpreted the results and prepared the manuscript. G.M. devised and directed the research and wrote the paper.

**Competing interests** The other authors declare no competing interests.

**Additional information**
**Correspondence and requests for materials** should be addressed to Gero Miesenböck.

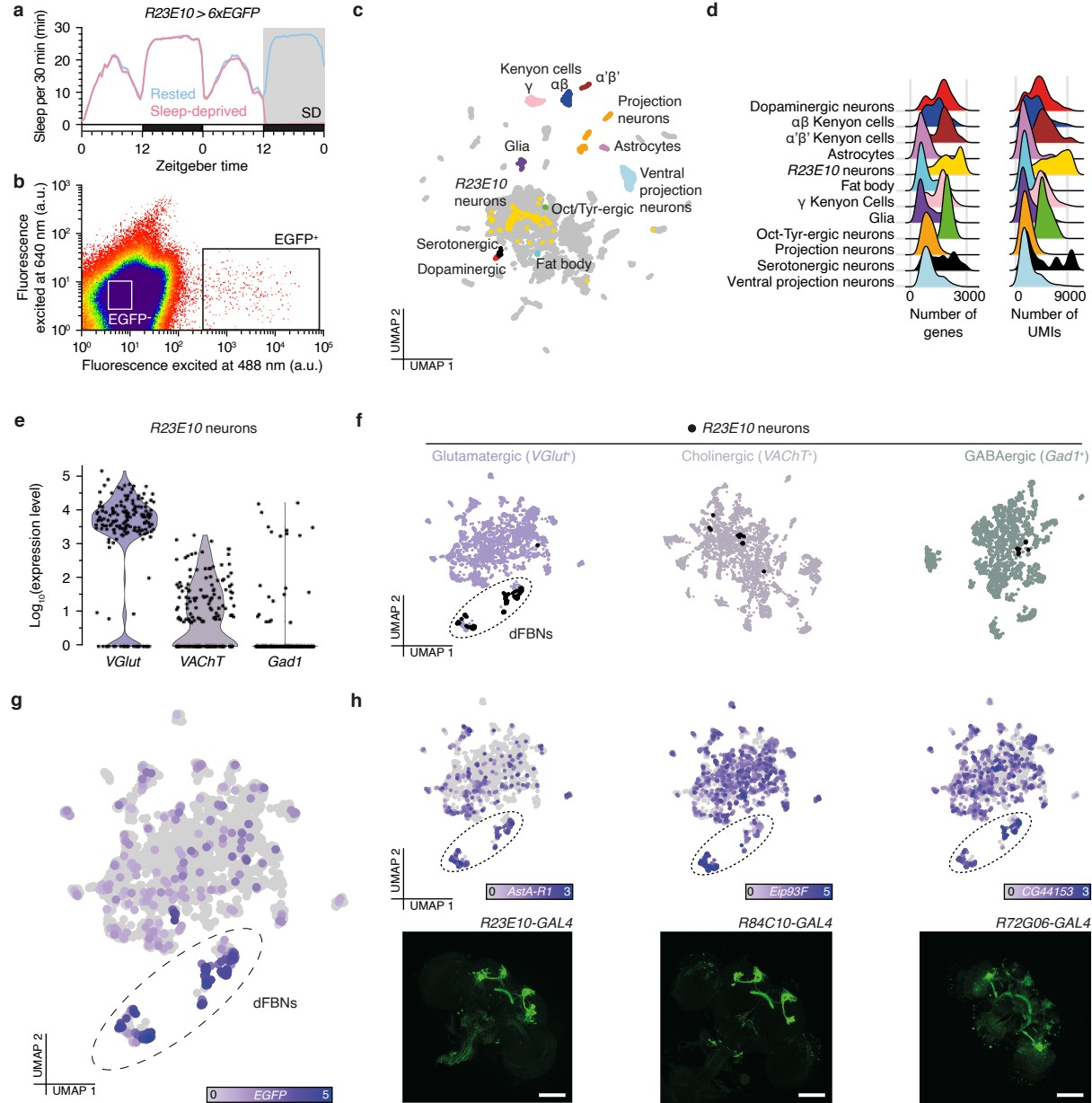

**Extended Data Fig. 1 | Identification of dFBNs and their transcriptomic response to sleep deprivation by single-cell RNA-sequencing. a**, Sleep profiles of flies expressing *R23E10-GAL4*-driven 6xEGFP under control (blue, *n* = 558) and sleep deprivation (SD) conditions (red, *n* = 432) before single-cell RNA sequencing. **b**, Pseudocolour plot of the gating strategy used to isolate EGFP-positive (0.03% of total) and EGFP-negative cells (14.64% of total) by flow cytometry. **c**, Uniform manifold approximation and projection (UMAP) representation of cells in the fly brain. Highlighted cell types, including neurons nominated by *R23E10-GAL4* (yellow), were identified as detailed in Methods. **d**, Distribution of the number of unique molecular identifiers (UMIs) and genes per cell for each annotated cell type. **e**, log-normalized distribution of the expression levels of markers for the fast-acting neurotransmitters glutamate, acetylcholine, and GABA in *R23E10-GAL4* neurons. **f**, *R23E10-GAL4* neurons (black) mapped onto re-clustered representations of cells expressing glutamatergic, cholinergic, and GABAergic markers. Bona fide dFBNs form a distinct glutamatergic cluster (dashed outline). **g**, **h**, Compared to the rest of the glutamatergic brain, dFBNs are enriched in EGFP (**g**) and transcripts of genes whose enhancer fragments label dFBNs in *GAL4* lines generated and imaged by the FlyLight project[58] (**h**). log-normalized expression levels are colour-coded according to the keys below each panel. Scale bars in **h**, 100 μm. Anatomical images in panel **h** reproduced from ref. 58, Cell Press, under a Creative Commons licence CC BY 4.0.

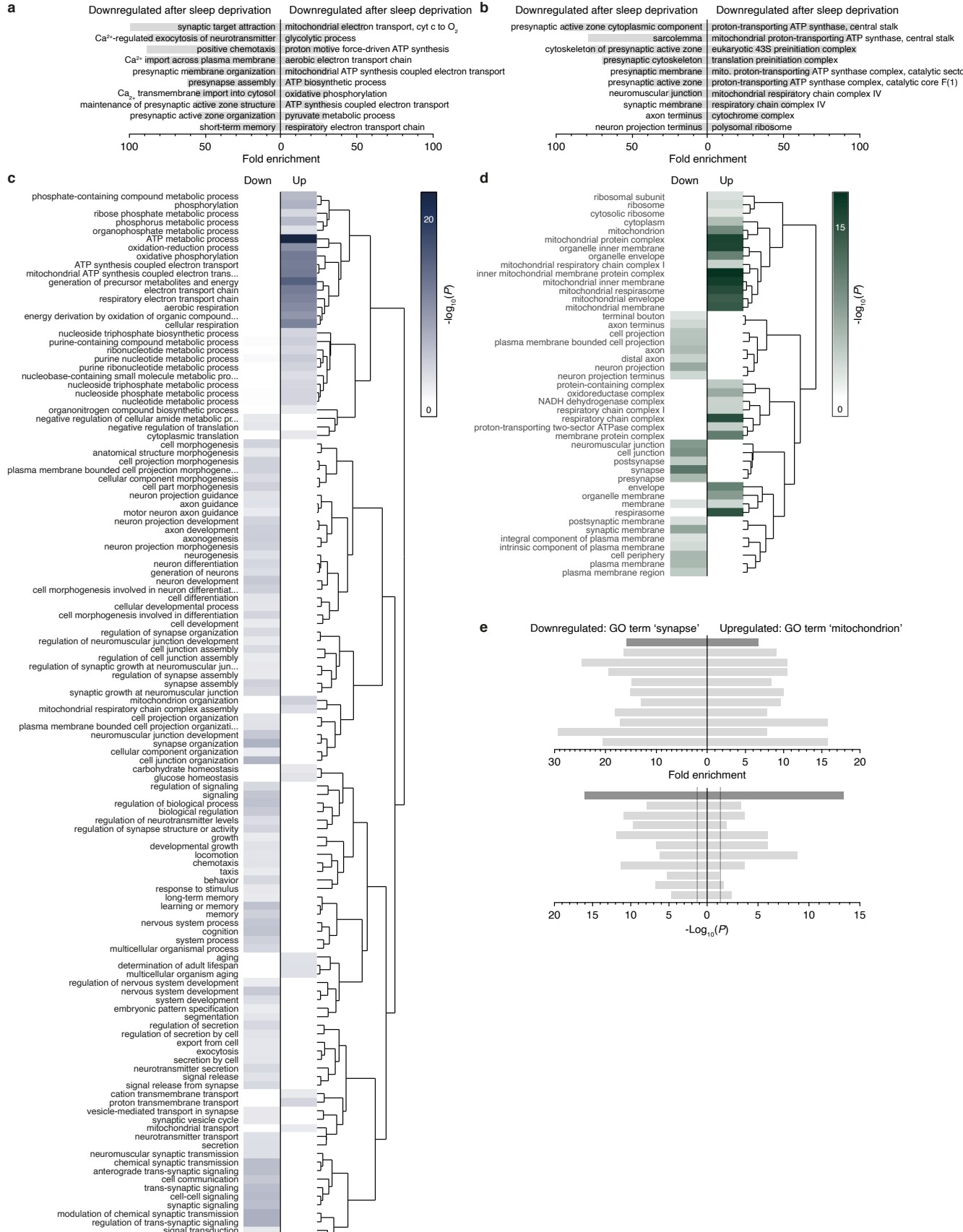

**Extended Data Fig. 2** | See next page for caption.

**Extended Data Fig. 2 | Gene ontology analysis of sleep history-dependent gene expression in dFBNs. a**–**d**, Enrichment of gene ontology (GO) 'biological process' (**a**, **c**) and 'cellular component' (**b**, **d**) terms in the set of genes whose expression in dFBNs varies with sleep history. Panels **a** and **b** plot the top ten enriched terms by PANTHER Overrepresentation Test (fold enrichments >100 are truncated). Panels **c** and **d** show heatmaps (computed with ViSEAGO and topGO) of GO terms attached to downregulated (left) and upregulated (right) differentially expressed genes. Dendrograms represent semantic groupings among GO terms. *P*-values are colour-coded according to the keys on the right. **e**, Enrichment (top) and uncorrected *P*-values (bottom, dotted lines at *P* = 0.05) of the 'cellular component' GO terms 'mitochondrion' and 'synapse' in the sets of up- and downregulated genes by PANTHER Overrepresentation Test, in the full data set (dark grey) and after randomly downsampling (light grey) the number of rested dFBN transcriptomes to the number of sleep-deprived dFBN transcriptomes (*n* = 86 cells in either condition). The downsampling process was repeated ten times using the 'max.cells.per.ident' argument of the 'FindMarkers' function in Seurat, with reproducible seedings from 1 to 10.

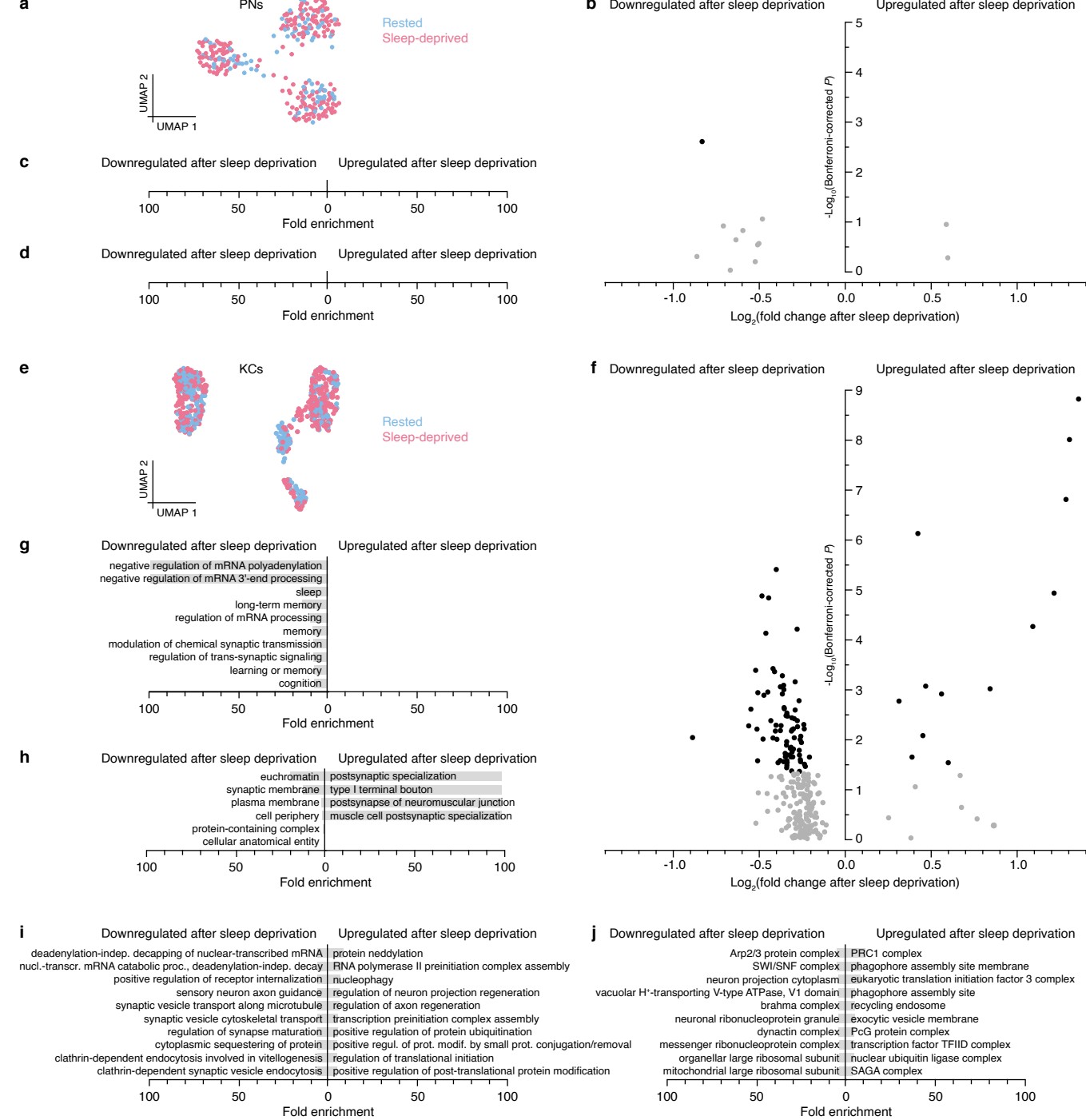

**Extended Data Fig. 3 | Sleep history-dependent gene expression in Kenyon cells, projection neurons and non-dFBN cells. a**, UMAP representation of projection neurons (PNs, *n* = 317) from rested (blue) and sleep-deprived brains (red) according to their gene expression profiles. **b**, Volcano plot of sleep history-dependent gene expression changes in PNs. A single signal with Bonferroni-corrected *P* < 0.05 (two-sided Wilcoxon rank-sum test) is indicated in black. **c**, **d**, PANTHER Overrepresentation Test fails to detect enriched 'biological process' (**g**) and 'cellular component' (**h**) gene ontology (GO) terms in the set of differentially expressed PN genes. **e**, UMAP representation of Kenyon cells (KCs, *n* = 603) from rested (blue) and sleep-deprived brains (red) according to their gene expression profiles. **f**, Volcano plot of sleep history-dependent gene expression changes in PNs. Signals with Bonferroni-corrected *P* < 0.05 (two-sided Wilcoxon rank-sum test) are indicated in black. **g**, **h**, Enrichment of the top ten downregulated and upregulated 'biological process' (**g**) and 'cellular component' (**h**) GO terms in the set of differentially expressed KC genes by PANTHER Overrepresentation Test (fold enrichments >100 are truncated). **i**, **j**, Enrichment of the top ten downregulated and upregulated 'biological process' (**i**) and 'cellular component' (**j**) GO terms in the set of genes with differential expression in all non-dFBN cells by PANTHER Overrepresentation Test.

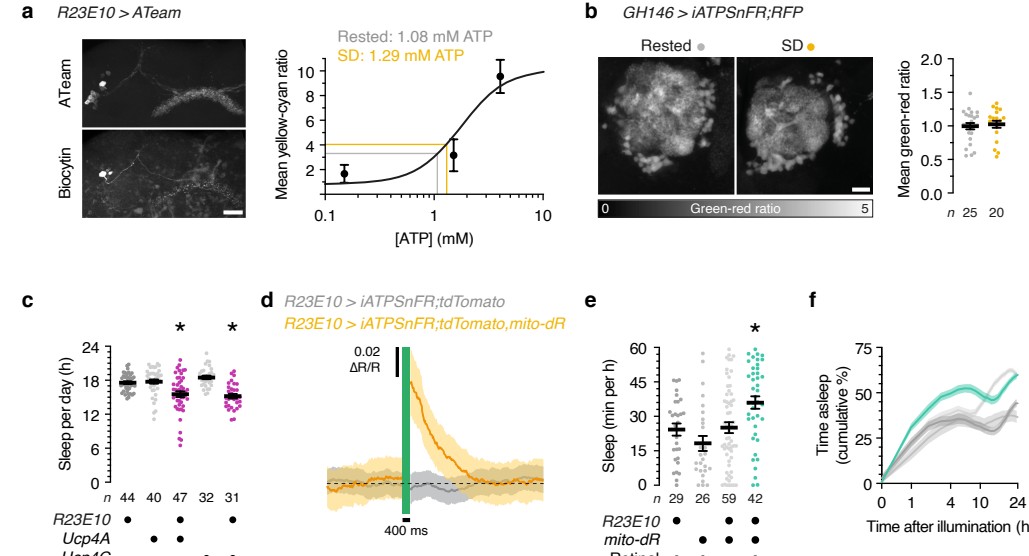

**Extended Data Fig. 4 | Relationships between ATP levels, mitochondrial protonmotive force, and sleep in PNs and dFBNs. a**, Maximum-intensity projection of dFBNs expressing *R23E10-GAL4*-driven ATeam (top left, CFP channel). One dFBN has been filled with biocytin and a defined ATP concentration through a whole-cell electrode (bottom left). The right panel shows the least-squares fit of a Hill equation (dissociation constant 1.75 mM ATP, Hill coefficient 2.1) to the mean yellow-to-cyan emission ratio of biocytin-labelled dFBNs containing 0.15, 1.5, and 4 mM ATP (*n* = 3 cells each). **b**, Summed-intensity projections of PN dendrites expressing iATPSnFR plus RFP, in rested and sleep-deprived (SD) flies. Emission ratios are intensity-coded according to the key below and unaltered by sleep deprivation (*P* = 0.6616, two-sided *t*-test). **c**, Sleep in flies expressing *R23E10-GAL4*-driven Ucp4A or Ucp4C and parental controls (*P* ≤ 0.0139, Dunn's test after Kruskal-Wallis ANOVA). **d**, A 400-ms

pulse of green light elevates ATP in dFBNs expressing iATPSnFR plus tdTomato and mito-dR but not in dFBNs lacking mito-dR (*n* = 5 flies of either genotype, *Δp* photogeneration effect: *P* < 0.0001, time × *Δp* photogeneration interaction: *P* < 0.0001, two-way ANOVA). **e, f**, Sleep during the first 60 min after illumination (**e**, *P* ≤ 0.0279, Dunn's test after Kruskal-Wallis ANOVA) and cumulative sleep percentages in flies expressing *R23E10-GAL4*-driven mito-dR, with or without retinal, and parental controls (**f**, *Δp* photogeneration effect: *P* < 0.0001, time × *Δp* photogeneration interaction: *P* < 0.0001, mixed-effects model). Asterisks, significant differences (*P* < 0.05) from both parental controls or in planned pairwise comparisons. Data are means ± s.e.m.; *n*, number of cells (**a**), antennal lobe glomeruli (**b**), or flies (**c**–**f**). Scale bars, 20 μm (**a, b**). For statistical details see Supplementary Table 2.

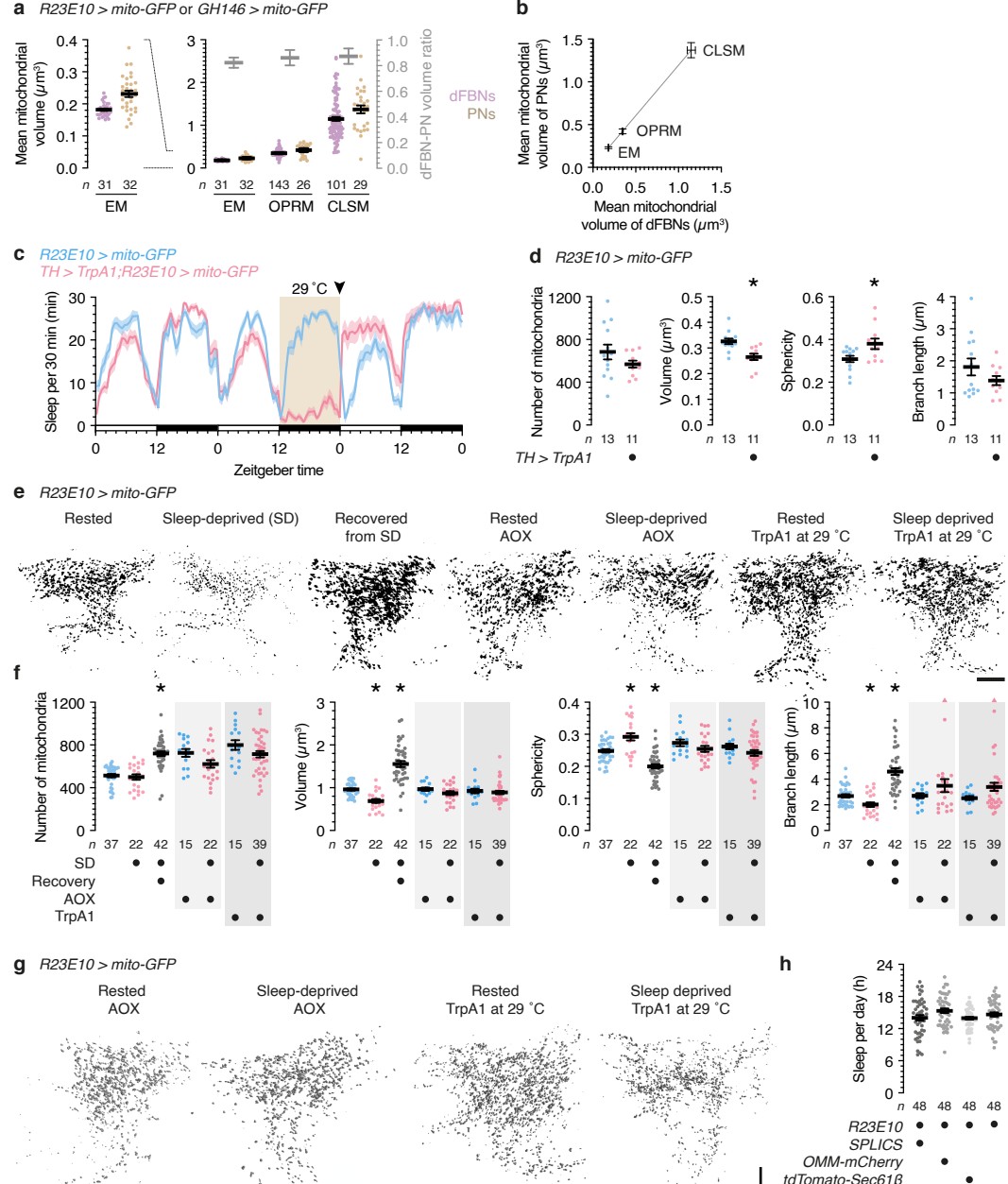

**Extended Data Fig. 5 | Sleep history alters the morphology of dFBN mitochondria. a**, Mean volumes (left y-axes) and volume ratios (right y-axis) of dFBN and PN mitochondria determined by volume electron microscopy (EM), optical photon reassignment microscopy (OPRM), and confocal laser-scanning microscopy (CLSM). **b**, Correlation between dFBN and PN mitochondrial volume estimates obtained by EM, OPRM, and CLSM (residual s.d. = 0.0041). EM data are from the hemibrain connectome[39]; OPRM and CLSM measurements of mitochondrial volumes in flies expressing mito-GFP are re-plotted from Fig. 3b and Extended Data Fig. 6b, and from **f** and Extended Data Fig. 6d, respectively. **c**, **d**, Sleep deprivation via thermogenetic activation of arousing dopaminergic neurons (**c**) causes mitochondrial fragmentation detected by OPRM (**d**, number of mitochondria: $P = 0.1812$, volume: $P = 0.0010$, sphericity: $P = 0.0192$, branch length: $P = 0.2013$, two-sided $t$-test). Experimental and control flies ($n = 30$ and 33, respectively) were reared and maintained at 21 °C and shifted to 29 °C between zeitgeber times 12 and 24 on day 2. The arrowhead marks the time point when 11 experimental and 13 control flies were removed and dissected for mitochondrial morphometry. **e**, **f**, Maximum intensity projections (**e**) and morphometric parameters (**f**) of automatically detected mitochondria in CLSM image stacks of dFBN dendrites in rested flies, sleep-deprived flies, flies allowed to recover for 24 h after sleep deprivation, and rested and sleep-deprived flies co-expressing *R23E10-GAL4*-driven AOX or TrpA1, which was activated at 29 °C.

Sleep history-dependent changes in mitochondrial volume ($P = 0.0025$, Holm-Šídák test after ANOVA), sphericity ($P = 0.0001$, Holm-Šídák test after ANOVA), and branch length ($P = 0.0414$, Holm-Šídák test after ANOVA) are occluded by the co-expression of AOX ($P \geq 0.1515$, two-sided $t$- or Mann-Whitney test) or the simultaneous activation of TrpA1 ($P \geq 0.2002$, two-sided $t$- or Mann-Whitney test) and overcorrected after recovery sleep (all parameters: $P < 0.0001$, Holm-Šídák test after ANOVA). The number of mitochondria is unchanged by sleep deprivation ($P > 0.9999$) but elevated after recovery sleep ($P < 0.0001$, Dunn's test after Kruskal-Wallis ANOVA). Two data points exceeding the y-axis limits are plotted as triangles at the top of the graphs; mean and s.e.m. are based on the actual values. **g**, Volumetric renderings of automatically detected mitochondria in OPRM image stacks of dFBN dendrites in rested and sleep-deprived flies co-expressing *R23E10-GAL4*-driven AOX or TrpA1, which was activated at 29 °C. **h**, Sleep in flies expressing *R23E10-GAL4*-driven split-GFP-based contact site sensors (SPLICS) or fluorescent fusion proteins located in the outer mitochondrial (OMM), endoplasmic reticulum (Sec61β), or plasma membrane (CD4) ($P = 0.0648$, ANOVA). Data are means ± s.e.m. or ratios of means ± error-propagated s.e.m. (**a**, light gray); *n*, number of cells (**a**, **b**, EM), dendritic fields (**a**, **b**, OPRM and CLSM, **d**, **f**), or flies (**c**, **h**); asterisks, significant differences ($P < 0.05$) in planned pairwise comparisons. Scale bars, 10 μm (**e**, **g**). For statistical details see Supplementary Table 2.

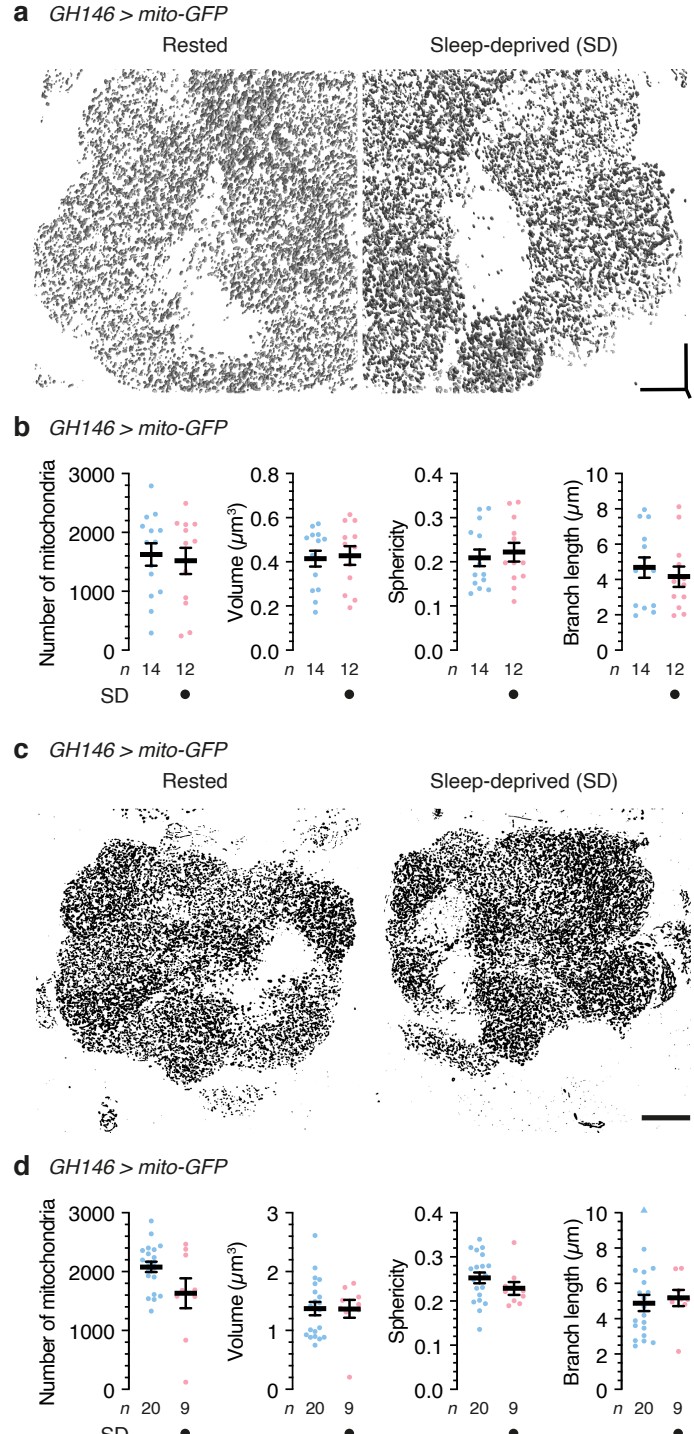

**Extended Data Fig. 6 | Sleep history does not alter the morphology of PN mitochondria. a**, **b**, Volumetric renderings (**a**) and morphometric parameters (**b**) of automatically detected mitochondria in OPRM image stacks of PN dendrites in rested and sleep-deprived flies. Mitochondrial number ($P = 0.7077$, two-sided $t$-test), volume ($P = 0.8074$, two-sided $t$-test), sphericity ($P = 0.6500$, two-sided $t$-test), and branch length ($P = 0.5326$, two-sided $t$-test) are unaffected by sleep deprivation. **c**, **d**, Maximum intensity projections (**c**) and morphometric parameters (**d**) of automatically detected mitochondria in CLSM image stacks of PN dendrites in rested and sleep-deprived flies. Mitochondrial number ($P = 0.2534$, two-sided Mann-Whitney test), volume ($P = 0.7637$, two-sided Mann-Whitney test), sphericity ($P = 0.1953$, two-sided Mann-Whitney test), and branch length ($P = 0.6972$, two-sided $t$-test) are unaffected by sleep deprivation. One data point exceeding the y-axis limits is plotted as a triangle at the top of the right-hand graph; mean and s.e.m. are based on the actual values. Scale bars, 10 μm (**a**), 20 μm (**c**). For statistical details see Supplementary Table 2.

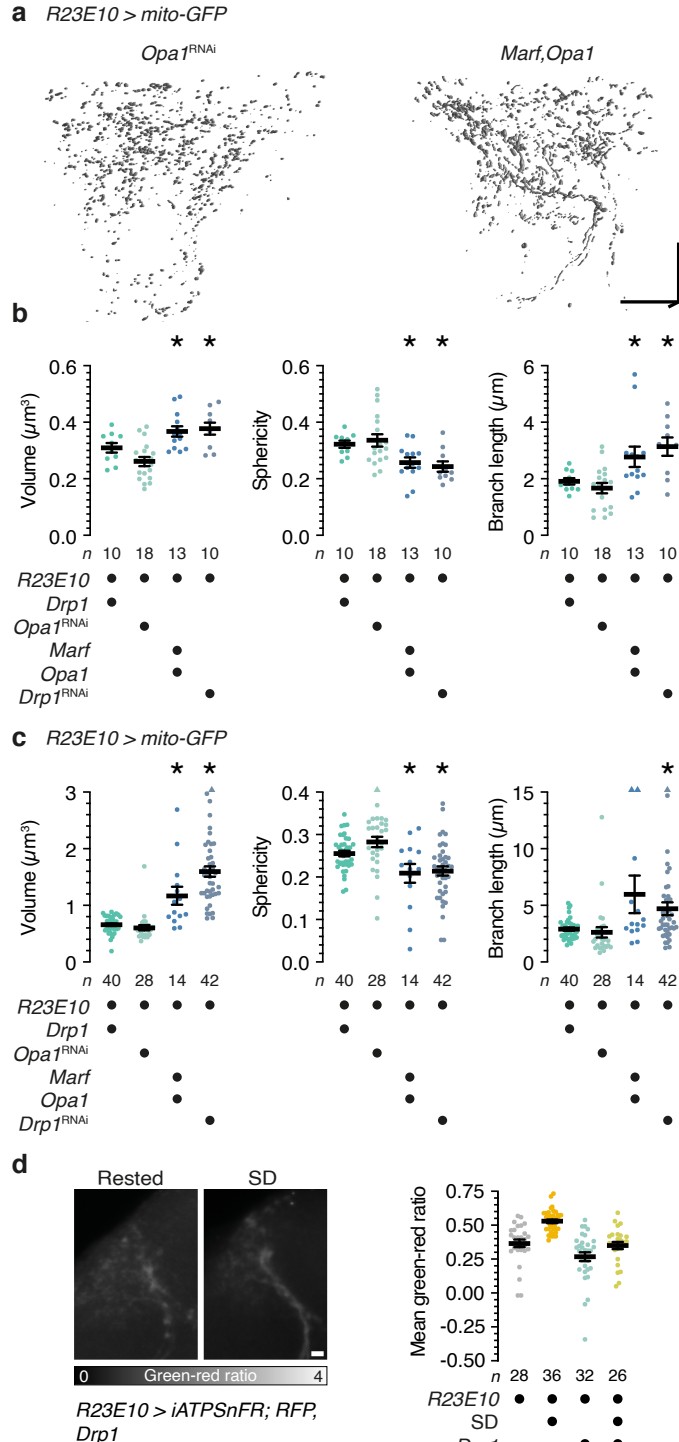

**Extended Data Fig. 7 | Morphological and metabolic consequences of inducing mitochondrial fission or fusion in dFBNs. a**, **b**, Volumetric renderings (**a**) and morphometric parameters (**b**) of automatically detected mitochondria in OPRM image stacks of dFBN dendrites. Flies carried *R23E10-GAL4*-driven overexpression constructs or RNAi transgenes targeting mitochondrial fission or fusion machinery. Manipulations that increase fission (green) or fusion (blue) have opposite effects on mitochondrial volume ($P \leq 0.0480$, Holm-Šídák test after ANOVA), sphericity ($P \leq 0.0344$, Holm-Šídák test after ANOVA), and branch length ($P \leq 0.0326$, Holm-Šídák test after ANOVA). **c**, Morphometric parameters of automatically detected mitochondria in CLSM image stacks of dFBN dendrites. Flies carried *R23E10-GAL4*-driven overexpression constructs or RNAi transgenes targeting mitochondrial fission or fusion machinery. Manipulations that increase fission (green) or fusion

(blue) have opposite effects on mitochondrial volume ($P \leq 0.0062$, Dunn's test after Kruskal-Wallis ANOVA), sphericity ($P \leq 0.0170$, Holm-Šídák test after ANOVA), and branch length ($P \leq 0.0427$, Dunn's test after Kruskal-Wallis ANOVA). Five data points exceeding the y-axis limits are plotted as triangles at the top of the graphs; mean and s.e.m. are based on the actual values. **d**, Summed-intensity projections of dFBN dendrites expressing Drp1 and iATPSnFR plus RFP, in rested and sleep-deprived (SD) flies. Emission ratios are intensity-coded according to the key below and reduced in dFBNs expressing Drp1, irrespective of sleep history (Drp1 effect: $P < 0.0001$, sleep history effect: $P < 0.0001$, Drp1 × sleep history interaction: $P = 0.1112$; two-way ANOVA). Data are means ± s.e.m.; *n*, number of dendritic fields; asterisks, significant differences ($P < 0.05$) from both manipulations increasing fission. Scale bars, 10 μm (**a**), 5 μm (**d**). For statistical details see Supplementary Table 2.

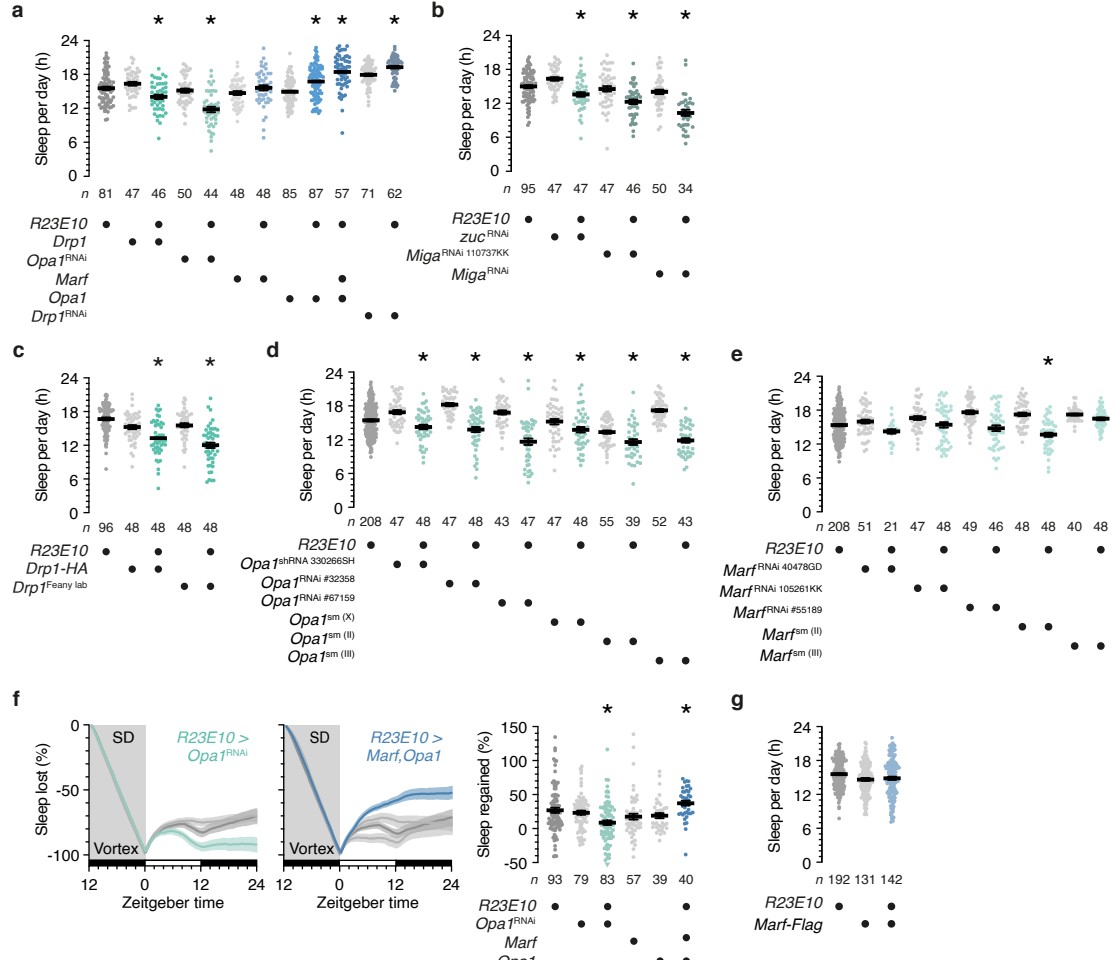

**Extended Data Fig. 8 | Inducing mitochondrial fission or fusion in dFBNs alters sleep. a**, **b**, Sleep in flies expressing *R23E10-GAL4*-driven fission or fusion proteins, or RNAi transgenes targeting transcripts encoding these proteins (**a**) or proteins regulating phosphatidic acid levels (**b**), and their parental controls. With the exception of the overexpression of Marf alone ($P \geq 0.1622$, Holm-Šídák test after ANOVA), manipulations that increase fission (green) or fusion (blue) alter sleep in opposite directions (GTPases: $P \leq 0.0115$, phosphatidic acid regulators: $P \leq 0.0381$; Holm-Šídák test after ANOVA). **c**–**e**, Sleep in flies carrying *R23E10-GAL4*-driven Drp1 overexpression constructs or RNAi transgenes targeting mitochondrial fusion proteins not included in **a**, **b** and Fig. 4c: two independent constructs for Drp1 overexpression (**c**, $P \leq 0.0209$, Dunn's test after ANOVA); six independent RNAi transgenes directed against Opa1 (**d**, $P \leq 0.0199$, Holm-Šídák test after ANOVA); and five independent RNAi transgenes directed

against Marf (**e**, $P \geq 0.1017$ relative to ≥1 parental control with the exception of *R23E10 > Marf*^sm(II)^, Holm-Šídák test after ANOVA). **f**, Manipulations that increase fission (*R23E10-GAL4 > Opa1*^RNAi^, green) or fusion (*R23E10 > Marf,Opa1*, blue) alter the time courses (left panels, genotype effects: $P \leq 0.0213$, time × genotype interactions: $P < 0.0001$, two-way repeated-measures ANOVA) and percentages of sleep rebound after deprivation (SD) in opposite directions (right panel, genotype effect: $P \leq 0.0186$, Dunn's test after ANOVA). One data point exceeding the y-axis limits is plotted as a triangle at the bottom of the right-hand graph; mean and s.e.m. are based on the actual values. **g**, Sleep in flies carrying an *R23E10-GAL4*-driven Marf overexpression construct not included in **a** and Fig. 4c ($P \geq 0.1252$, Dunn's test after Kruskal-Wallis ANOVA). Data are means ± s.e.m.; *n*, number of flies; asterisks, significant differences ($P < 0.05$) from both parental controls. For statistical details see Supplementary Table 2.

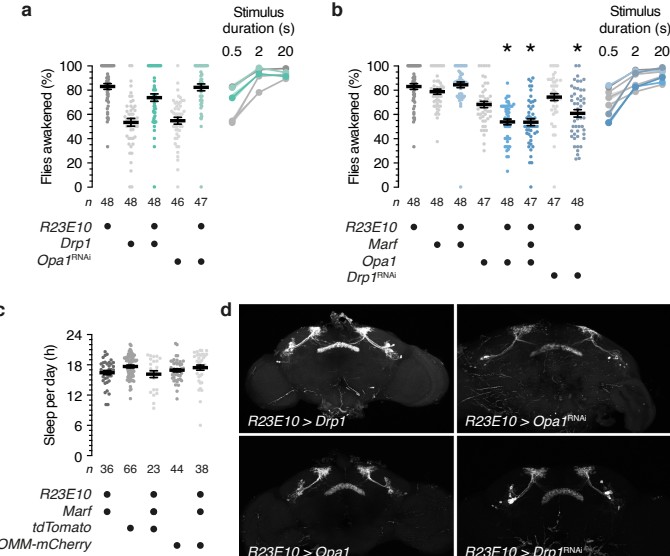

**Extended Data Fig. 9 | Inducing mitochondrial fission or fusion in dFBNs alters arousal thresholds without causing overexpression artefacts or anatomical defects. a**, **b**, Percentages of flies awakened by mechanical stimuli lasting 0.5 s (left panels), 2 s, or 20 s (right panels). The average percentages awakened by 0.5-s stimuli in the left panels are reproduced on the right. Manipulations that increase fission fail to lower the arousal threshold, possibly because of a floor effect linked to the *R23E10-GAL4* strain (**a**, *P* ≥ 0.3354 relative to ≥1 parental control, Dunn's test after Kruskal-Wallis ANOVA). With the exception of the overexpression of Marf alone (*P* > 0.9999), manipulations that increase fusion raise the arousal threshold (**b**, *P* ≤ 0.0371, Dunn's test after Kruskal-Wallis ANOVA). **c**, Sleep in flies expressing *R23E10-GAL4*-driven

Marf is insensitive to the co-expression of fluorescent proteins in the cytoplasm (tdTomato) or the outer mitochondrial membrane (OMM-mCherry) (*P* ≥ 0.1106, Dunn's test after Kruskal-Wallis ANOVA), in contrast to the synergistic effect of overexpressing Opa1 (Fig. 4c, Extended Data Fig. 8a). **d**, Maximum-intensity projections of dFBNs in flies carrying *R23E10-GAL4*-driven overexpression constructs or RNAi transgenes targeting the mitochondrial fission or fusion machinery. Five brains per genotype were imaged; representative examples are shown. Data are means ± s.e.m.; *n*, number of flies; asterisks, significant differences (*P* < 0.05) from both parental controls. Scale bar, 100 μm (**d**). For statistical details see Supplementary Table 2.

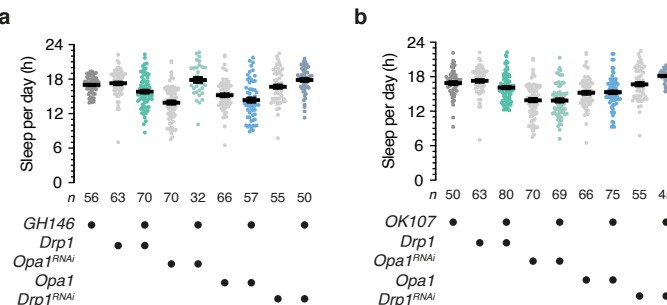

**Extended Data Fig. 10 | Inducing mitochondrial fission or fusion in PNs or KCs has no effect on sleep. a**, Sleep in flies carrying *GH146-GAL4*-driven overexpression constructs or RNAi transgenes targeting the mitochondrial fission or fusion machinery in PNs ($P \geq 0.0842$ relative to $\geq 1$ parental control, Holm-Šídák test after ANOVA). **b**, Sleep in flies carrying *OK107-GAL4*-driven overexpression constructs or RNAi transgenes targeting the mitochondrial fission or fusion machinery in KCs ($P \geq 0.0660$ relative to $\geq 1$ parental control, Holm-Šídák test after ANOVA). Data are means ± s.e.m.; *n*, number of flies. For statistical details see Supplementary Table 2.

# Reporting Summary

## Statistics

For all statistical analyses, confirm that the following items are present in the figure legend, table legend, main text, or Methods section.

| n/a | Confirmed | |
|---|---|---|
| ☐ | ☒ | The exact sample size (*n*) for each experimental group/condition, given as a discrete number and unit of measurement |
| ☐ | ☒ | A statement on whether measurements were taken from distinct samples or whether the same sample was measured repeatedly |
| ☐ | ☒ | The statistical test(s) used AND whether they are one- or two-sided<br>*Only common tests should be described solely by name; describe more complex techniques in the Methods section.* |
| ☒ | ☐ | A description of all covariates tested |
| ☐ | ☒ | A description of any assumptions or corrections, such as tests of normality and adjustment for multiple comparisons |
| ☐ | ☒ | A full description of the statistical parameters including central tendency (e.g. means) or other basic estimates (e.g. regression coefficient) AND variation (e.g. standard deviation) or associated estimates of uncertainty (e.g. confidence intervals) |
| ☐ | ☒ | For null hypothesis testing, the test statistic (e.g. *F*, *t*, *r*) with confidence intervals, effect sizes, degrees of freedom and *P* value noted<br>*Give P values as exact values whenever suitable.* |
| ☒ | ☐ | For Bayesian analysis, information on the choice of priors and Markov chain Monte Carlo settings |
| ☒ | ☐ | For hierarchical and complex designs, identification of the appropriate level for tests and full reporting of outcomes |
| ☒ | ☐ | Estimates of effect sizes (e.g. Cohen's *d*, Pearson's *r*), indicating how they were calculated |

*Our web collection on statistics for biologists contains articles on many of the points above.*

## Software and code

Policy information about availability of computer code

| Data collection | Sleep behaviour data were collected using the DAM system (Trikinetics) or a MWP Z2S unit (Zantiks) .<br>Two-photon imaging data were collected using ScanImage 5.4.0, running on MATLAB2015a.<br>Confocal imaging data were collected using the Leica LAS AF 2.7.3.9723 or ZEN blue 3.3.<br>Super-resolution imaging data were collected using the Olympus cellSens Dimension 4.3.1 platform.<br>Electrophysiology data were acquired with custom protocols in pCLAMP 11.2 (Molecular Devices). |
|---|---|
| Data analysis | ScRNA-seq data were aligned and annotated according to James Nemesh, McCarroll Lab drop-seq core computational protocol V2.0.0 and analysed in R using Seurat v4.1.<br>Light microscopy data were analysed using existing, adapted, or newly developed (semi-)automated routines in Fiji 2.14.0/1.54f, using the DeconvolutionLab2 2.1.2 and Mitochondria Analyzer 2.3.1 plugins where indicated.<br>Two-photon imaging data were analysed in MATLAB 2023a.<br>Electrophysiological data were analysed using version 3.0c of the NeuroMatic package in Igor Pro 8.04 (WaveMetrics)..<br>Sleep behaviour data were analysed with the Sleep and Circadian Analysis MATLAB Program (SCAMP v3) or a custom MATLAB script for video tracking.<br>Gene ontologies were computed in PANTHER v17 or the ViSEAGO 1.4.0 and topGO 2.42.0 packages.<br>Behavioural, imaging, and electrophysiological data were analysed in Prism 10 (GraphPad) and SPSS Statistics 29 (IBM). |

For manuscripts utilizing custom algorithms or software that are central to the research but not yet described in published literature, software must be made available to editors and reviewers. We strongly encourage code deposition in a community repository (e.g. GitHub). See the Nature Portfolio guidelines for submitting code & software for further information.

# Data

Policy information about <u>availability of data</u>

All manuscripts must include a <u>data availability statement</u>. This statement should provide the following information, where applicable:
- Accession codes, unique identifiers, or web links for publicly available datasets
- A description of any restrictions on data availability
- For clinical datasets or third party data, please ensure that the statement adheres to our <u>policy</u>

Single-cell transcriptomic data were aligned to the Drosophila melanogaster genome release BDGP6.22 and can be found in NCBI's Gene Expression Omnibus repository under accession number GSE256379. All other data generated and analysed in this study are included in the Source Data.

# Research involving human participants, their data, or biological material

Policy information about studies with <u>human participants or human data</u>. See also policy information about <u>sex, gender (identity/presentation), and sexual orientation</u> and <u>race, ethnicity and racism</u>.

| | |
|---|---|
| Reporting on sex and gender | n/a |
| Reporting on race, ethnicity, or other socially relevant groupings | n/a |
| Population characteristics | n/a |
| Recruitment | n/a |
| Ethics oversight | n/a |

Note that full information on the approval of the study protocol must also be provided in the manuscript.

# Field-specific reporting

Please select the one below that is the best fit for your research. If you are not sure, read the appropriate sections before making your selection.

☒ Life sciences          ☐ Behavioural & social sciences          ☐ Ecological, evolutionary & environmental sciences

For a reference copy of the document with all sections, see <u>nature.com/documents/nr-reporting-summary-flat.pdf</u>

# Life sciences study design

All studies must disclose on these points even when the disclosure is negative.

| | |
|---|---|
| Sample size | Sample sizes are provided in each figure and extended data figure or its legend.<br>Sample sizes in behavioural experiments were chosen to detect 2-h differences in daily sleep with a power of 0.8.<br>Sample sizes in electrophysiological experiments are based on precedent (Kempf et al., Nature 2019). |
| Data exclusions | In single-cell transcriptomics, genes detected in fewer than 3 cells were excluded, and only cells associated with 800–10,000 UMIs and 200–5,000 transcripts were analysed.<br>Immobile flies (< 2 beam breaks per 24 h) in standard sleep assays in the Trikinetics Drosophila Activity Monitor system were excluded from the analysis.<br>In sleep analyses in the Zantiks unit, immobile flies (>98% zero-speed bins during ≥2 consecutive hours until the end of the recording) were excluded, beginning with the hour preceding the onset of immobility.<br>Only flies losing >95% of baseline sleep were included in measurements of sleep rebound after deprivation.<br>Only anatomically intact specimens from live flies (at the point of dissection) were used for mitochondrial morphometry.<br>Only dFBNs firing more than one action potential in response to depolarizing current injections, with resting potentials <−30 mV and series resistances <50 MΩ, were characterized. |
| Replication | All behavioural and imaging experiments were run at least three times, on different days and with different batches of flies. Behavioural and morphometric analyses were replicated in independent series of experiments, using two different dFBN-targeting GAL4 drivers and two imaging methods .All replicates are included in figures and extended data figures; only Extended Data Fig. 9d shows representative examples. |
| Randomization | Female flies of a given genotype, as indicated in Methods and figure legends, were randomly selected for analysis. Controls and experimental groups were always tested in parallel and in randomized order. |
| Blinding | The investigators were blind to group allocation in imaging experiments but not otherwise. Measurements and analyses were automated. |

# Reporting for specific materials, systems and methods

We require information from authors about some types of materials, experimental systems and methods used in many studies. Here, indicate whether each material, system or method listed is relevant to your study. If you are not sure if a list item applies to your research, read the appropriate section before selecting a response.

## Materials & experimental systems

| n/a | Involved in the study |
|---|---|
| ☐ | ☒ Antibodies |
| ☒ | ☐ Eukaryotic cell lines |
| ☒ | ☐ Palaeontology and archaeology |
| ☐ | ☒ Animals and other organisms |
| ☒ | ☐ Clinical data |
| ☒ | ☐ Dual use research of concern |
| ☒ | ☐ Plants |

## Methods

| n/a | Involved in the study |
|---|---|
| ☒ | ☐ ChIP-seq |
| ☐ | ☒ Flow cytometry |
| ☒ | ☐ MRI-based neuroimaging |

## Antibodies

| | |
|---|---|
| Antibodies used | Primary: Mouse anti-DDK, (AB_2622345, clone OTI4C5, F-tag-01 – TA100011), OriGene (1:1000). Secondary: Goat anti-Mouse Alexa 633, (AB_2535719), Thermo Fisher Scientific (1:500). |
| Validation | The primary antibody was validated by the manufacturer in mammalian cells transfected with DDK-tagged vectors in Western blots and by immunofluorescence microscopy. A previous publication of ours validated the antibody in Drosophila (Rorsman et al., Nature 2025). The secondary antibody was validated by the manufacturer in mammalian cells stained with mouse monoclonal primary antibody and by us in Drosophila (Rorsman et al., Nature 2025). |

## Animals and other research organisms

Policy information about [studies involving animals](); [ARRIVE guidelines]() recommended for reporting animal research, and [Sex and Gender in Research]()

| | |
|---|---|
| Laboratory animals | Transgenic Drosophila melanogaster strains; genotypes indicated in Methods and Supplementary Table 3. Females aged 2–6 days after eclosion, as indicated in Methods, were used in experiments. |
| Wild animals | No wild animals were used in this study. |
| Reporting on sex | Female flies were used in all experiments because of their larger body size. |
| Field-collected samples | No field-collected samples were used in this study. |
| Ethics oversight | No ethical approval was required for research on Drosophila melanogaster. |

Note that full information on the approval of the study protocol must also be provided in the manuscript.

## Plants

| | |
|---|---|
| Seed stocks | *Report on the source of all seed stocks or other plant material used. If applicable, state the seed stock centre and catalogue number. If plant specimens were collected from the field, describe the collection location, date and sampling procedures.* |
| Novel plant genotypes | *Describe the methods by which all novel plant genotypes were produced. This includes those generated by transgenic approaches, gene editing, chemical/radiation-based mutagenesis and hybridization. For transgenic lines, describe the transformation method, the number of independent lines analyzed and the generation upon which experiments were performed. For gene-edited lines, describe the editor used, the endogenous sequence targeted for editing, the targeting guide RNA sequence (if applicable) and how the editor was applied.* |
| Authentication | *Describe any authentication procedures for each seed stock used or novel genotype generated. Describe any experiments used to assess the effect of a mutation and, where applicable, how potential secondary effects (e.g. second site T-DNA insertions, mosiacism, off-target gene editing) were examined.* |

# Flow Cytometry

## Plots

Confirm that:

☒ The axis labels state the marker and fluorochrome used (e.g. CD4-FITC).

☒ The axis scales are clearly visible. Include numbers along axes only for bottom left plot of group (a 'group' is an analysis of identical markers).

☒ All plots are contour plots with outliers or pseudocolor plots.

☒ A numerical value for number of cells or percentage (with statistics) is provided.

## Methodology

| | |
|---|---|
| Sample preparation | Freshly dissected fly brains were dissociated enzymatically and mechanically and filtered as described in Methods. Dead cells were excluded with the help a DAPI viability dye (1 μg ml-1, BD Pharmingen). |
| Instrument | MoFlo Astrios (Beckman Coulter) or FACSAria III (Becton Dickinson) |
| Software | FACSDiva (Becton Dickinson) |
| Cell population abundance | Approximately 15 EGFP-positive cells per dissected brain at a concentration of 300 cells/μl. Purity was ascertained visually by fluorescence microscopy. |
| Gating strategy | Single cells were gated for based on forward and side scatter parameters of wild-type fly brains, followed by subsequent gating for EGFP-positive and EGFP-negative cells (Extended Data Fig. 1b). |

☒ Tick this box to confirm that a figure exemplifying the gating strategy is provided in the Supplementary Information.

