## [Peer Review File · Nature]

Mitochondrial origins of the pressure to sleep

Corresponding Author: Professor Gero Miesenboeck

Version 0:

Reviewer comments:

Referee #1

(Remarks to the Author)

In this manuscript, Sarnataro and colleagues from the Miesenböck laboratory use *D. melanogaster* as a model to explore the biological basis of sleep pressure. Using a range of tools from single-cell transcriptomics to in vivo imaging and sleep measurements they show that mitochondrial dynamics and bioenergetics in dorsal fan-shaped body (dFBNs) are involved in determining the sleep pressure after deprivation.

This is an exciting claim. But I have two major concerns – first, for the non-sleep or fly experts (to which I belong), it is not immediately clear how this study fundamentally moves beyond the Miesenböck's prior paper entitled "A potassium channel β -subunit couples mitochondrial electron transport to sleep" from 2019, which uses similar tools in similar contexts, and in my reading already supports a view that there is an immediate link between mitochondrial respiration and sleep. I understand that the new manuscript (together with several others that are submitted in parallel) adds a lot more detail and claims to have discovered the biological substrate of sleep pressure (namely the ATP level and the corresponding redox equilibria of the respiratory chain), further linking this to the mitochondrial fission/fusion balance in specific neurons. Eventually, this is an editorial decision, also based on considering the other submitted manuscripts by the same group. But I at least could not exactly discern how fundamentally novel the proposed concepts are, and I was surprised to see a whole segment in the Results (paragraph 2 first part of paragraph 3 on page 5) that in essence say that some key data were published prior or will be published elsewhere.

However, beyond this somewhat subjective assessment, I also think the manuscript makes many assumptions about the tools employed that need substantial further controls, as detailed below.

Specific concerns:

The authors use a plethora of tools to manipulate mitochondria and measure the effects on various aspects of sleep. My problem is that most of the tools are not verified in detail. I understand that the authors would argue that perhaps the multiple streams of evidence make their claims independent of any single tool performing exactly as assumed, but I still think this needs to be shown in the specific settings used here.

Examples are:

Fig.1 – The extent and specificity for mitochondrial genes of the transcriptome changes are impressive – but which part of this can be translated to the functionally relevant proteome of mitochondria during the relevant time period (which I assume is 12 hours). In other words, what changes in mitochondrial molecular composition actually result for the sleep-deprivation induced cellular transcriptome – or even in the natural sleep-wake cycle?

Fig 2b,c – The authors claim a "15-fold higher" ATP level in deprived neurons. I think this needs in situ calibration, which could be tricky.

Fig. 2e – Why is there no recovery after the illumination ceases – is this a channelrhodopsin that stays activated or is this a change in sleeping status?

Fig. 2g,h – Where is the mitochondrial channelrhodopsin verified in its specific action? I could only find a general reference in other biological contexts

Fig 3a,b – I have serious doubts that with the indicated techniques and work flow the morphology of mitochondria can be quantitatively captured – I know there is a reference to this workflow, but in my reading this is not established for flies or the specific imaging conditions used here. To control this, the authors should compare their data to published EM data, but ideally perform their own ultrastructural analysis also of the altered mitochondria, to prove that their read-out at the light microscopy level is valid for baseline and for their manipulations. I have seen data (e.g. in Justs et al., J Neurosci 2022) that seem to indicate that compared to mitochondria in other fly neurons, the volume estimates given here might be inflated (suggesting that the diffraction limit still determines much of what is measured). Please also explain, why the dendrites are the key compartment here – as I assume, somatic, or at least axonal mitochondria are not captured.

Fig. 3d – What is the ground truth here – I cannot determine the specificity of DRP translocation in these highly processed images.

Fig. 3e – Also here, EM evidence will be needed, and it needs to be shown that the contact sensor itself does not induce aggregation (it seems there is an effect outside the dFNB dendrites, see cluster to the lower right).

Fig. 3f – The provided image does not seem representative of (or even on the same scale as) the corresponding graphs

Fig. 4a,b: It needs to be shown (a) how the genetic manipulations affect protein levels in dFNBs and (b) how that affects mitochondrial morphology (this is the domain of EM or superresolution imaging, not confocal).

Minor comments

Overall, the referencing is sometimes very broad and should be limited to essential and primary literature, not general reviews, see p. 6 Ref 14-16.

The discussion is interesting, but perhaps a bit remote from the data – and should at least discuss relevant recent papers such as Haynes et al., Nat Neurosci 2024 (which might have appeared only after the authors completed this manuscript), and perhaps also discuss the question whether there are any hints in other organisms suggesting a similar basis for sleep pressure.

Referee #2

(Remarks to the Author)

The manuscript “Mitochondrial origins of the pressure to sleep” by Sarnataro et al. claims to demonstrate a central role of mitochondrial aerobic metabolism in generating the pressure to sleep. The hypothesis is novel and interesting and the authors bring to bear a slew of approaches and novel data to support the hypothesis. This work builds from the prior landmark paper (Kempf et al, 2019) showing that the buildup of mitochondrial-driven ROS activates sleep via the redox state of the K channel subunit Hyperkinetic. They extend these findings here by demonstrating wake-driven changes in mitochondrial gene transcripts and ROS driven mitochondrial fragmentation. A nice set of genetic/optogenetic perturbation experiments demonstrate the interplay of electron flux and ATP synthesis/levels is critical to dFBN activity in the control of sleep. The strength of the paper is in the elucidation of a molecular and cellular pathway linking energy availability and mitochondrial metabolism to the control of neurons that regulate sleep.

Major Comments:

1. While the paper makes use of many sophisticated techniques to substantiate their model, it is not clear how this paper on its own substantially advances the model originally put forth in Kempf (see Fig. 2 of that paper)
2. The paper is predicated on studies of the dFBN in sleep regulation. The authors in this manuscript make scant reference to two peer-reviewed studies (from the Joiner and Dissel groups) that have strongly questioned the preeminent role of the dFBN. From those studies it is clear that the 23E10 driver used widely in this study also drives expression in other potent sleep regulatory “bowtie” neurons. Thus the genetic/RNAi experiments (Fig. 2 and 4) using this driver ought to use a reagent that excludes a potential function in the “bowtie” neurons.
3. A major component of their model is the behavioral function of Hk in the dFNBs which was questioned recently. It would be good if this could be confirmed, addressed or refuted.
4. Haynes et al has shown a role for the mitochondrial fission protein Drp1 in sleep regulation—this paper extends that work to show its function in the dFBN (although with the caveat listed above). Nonetheless, it was difficult to discern the changes in mitochondrial morphology in Fig. 3. I wonder whether such changes require ultrastructural analyses to be more definitive. It is also not clear mechanistically how these changes are linked to cellular physiology important for sleep regulation.

Referee #3

(Remarks to the Author)

The manuscript by Sarnataro and colleagues presents a compelling set of experiments, in *Drosophila*, linking different aspects of mitochondrial biology with sleep regulation. The authors first use single-cell RNA sequencing to identify a population of dFNBs cells responsive to sleep deprivation (and/or stress). They then examine how manipulating different aspects of mitochondrial bioenergetics in this cell population influence sleep behavior on aggregate fly populations, showing that manipulation of mitochondrial dynamics genes influence sleep behavior in directions predicted by the model. The work appears to be of high quality and the question is of high importance. Some limitations of the work and opportunities to increase the impact of the manuscript are listed below.

In the scRNAseq data, there was a large difference in the number of dFNBs between the two groups (237 vs 86). Why is

this? This potentially undermines the power and introduced the potential of a skewed population of cells being studied between the groups. How can the authors rule out this possibility?

Given the different distribution of differentially expressed genes in Figure 1b, it may be useful to include non-significant datapoints (the bottom of the volcano). Having the x axis go from -1.5 to +1.5 would also help readers appreciate relative differences between Up and Down regulated groups of genes. This also applies to Extended Data figures.

The authors report that compared to rested, SD leads to a fifteen (15)-fold higher ATP levels. ATP levels tend to be strongly controlled and likely never vary that much in a living context. The datapoints also don't seem to show a 15-fold difference. Is this a mistake? Can the author clarify.

Minor related point: if the points were semi-transparent the reader could appreciate the density of datapoints within the distribution, which is obscured in the current version of the figures.

The authors should provide data confirming the extent to which their cell-type specific expression mutants are in fact cell type specific. How much leak is there outside the target cell type?

As presented, the mitochondrial morphology analysis is weak. The data for mitochondrial size and morphology appears to have been done using fairly low magnification. At low resolution with light microscopy, juxtaposed mitochondria (either post fission, or bound by inter-mitochondrial junctions, for example) will appear "fused" or longer than they actually are. The morphological alterations would be best demonstrated by transmission electron microscopy, or even more compellingly with serial scanning EM (FIB-SEM, SBF-SEM) where the morphology of mitochondria can be accurately determined.

The model presented suggests that sleep duration is influenced by the balance of mitochondrial fusion and fission. If this is correct and generalizable in *Drosophila*, as suggested by the experimental manipulations of the fission/fusion machinery, then this should explain why some flies sleep more while some sleep less. In the data from Figure 4b, it is clear that some control R23E10 flies are short sleepers (<10h per day) while others are long sleepers (>20h). If the author's model is correct and sleep behavior is under the control of mitochondria, it should be possible to predict the behavior of each fly, or of groups of very short, short, average, long, and very long sleeping flies, based on gene expression or some other measure of the mitochondrial system. Can the authors address the relevance of their model related to inter-individual differences within one or more populations of flies? If they could address this point in both reference and, for example, *Opa1*-RNAi flies, this would be very compelling and would increase the impact of the paper. Specifically, what is the proportion of variance (r^2 on a regression analysis) in sleep behavior that is accounted for by expression of fission or fusion genes, or the ratio of fission>fusion genes? Finding a positive result would increase the likelihood that this mechanism may have relevance to understand inter-individual differences in sleep in other animals. Finding a negative result (hypothesis true across groups, but not between individuals of a population) would not necessarily invalidate the relevance of the model, but would force an additional layer of interpretation that also could ultimately improve the paper.

I am not an electrophysiologist and have little understanding of the constraints associated with these experiments. That said, it seems that one additional approach to test the question above may be to relate the electrophysiological measures in Figure 4D with sleep behavior in individual flies. First establish how long a fly sleeps per day then measure electrophysiology on the same fly, and correlate these parameters to see how much variance is shared between these measurements. Are the electrophysiological properties in a fly sufficiently dominant and causal that they can be used to predict sleep behavior? Here the potential for bias is high, so these experiments should be done blinded where the person determining sleep duration creates a code (Fly A, B, C, D, ...) so the sleep duration data cannot include the electrophysiological recordings.

Is the approach to sleep deprive fly also "stressful"? How can the effects of sleep deprivation be disentangled from those of stress responses or adaptation?

References 2 and 5 are on BioRxiv. It would have helped to indicate as such, especially given that reference 5 is used to support some of this manuscript.

Version 1:

Reviewer comments:

Referee #1

(Remarks to the Author)

The authors have carefully and constructively addressed my concerns - thank you!

I appreciate their efforts to even address difficult challenges (such as the calibration of the ATP sensor), their explanation of unsurmountable technical problems (e.g. regarding my suggestions for EM analysis) and the clarification and correction of this that I either misunderstood or where I spotted minor mistakes.

So altogether, I am satisfied with the technical revisions made to address my concerns. For the question of novelty and the relationship to recent advances in the literature, I think the editorial team has to decide - but I would assume the invitation of a revision to the authors and a re-review to the referees, underscores the fact that editorially this manuscript is seen

favorably.

I think it is an impressive piece of work that will stimulate the growing field that ascribes systemic effects to neuronal mitochondria. I would be excited to see this work in print!

Referee #2

(Remarks to the Author)

The authors have made significant revisions addressing some concerns. The major issue that remains is the novelty of this paper relative to their prior publication Kempf et al. In the Fig. 2 legend of that paper the authors state, "When more electrons enter the transport chain than are needed for ATP synthesis, a backlog accumulates in the Q pool." This appears to be conceptually very close to the "imbalance" model that is presented in this paper, albeit with more experimental evidence .

The authors have also added important new data to strengthen the manuscript including adding refined split GAL4 drivers to exclude bowtie VNC-SP neurons in their key genetic experiments. It is noteworthy that some of the functional effects (e.g., Fig. 4C) are relatively modest with relatively large sample sizes. That being said, it is clear that the dFBNs do have a sleep regulatory role and that perturbations of mitochondrial fission and fusion do contribute to this function. In addition, the authors have gone to great lengths and made significant progress on substantiating the morphological changes using an optical super-resolution method. Collectively the mitochondrial fission/fusion role in sleep homeostasis is an advance, although with the caveat that Haynes et al had described the sleep function of the mitochondrial fission protein Drp1.

Referee #3

(Remarks to the Author)

The manuscript is significantly improved and all major points of concern have been addressed.

Three suggestions:

1. A short, lucid paragraph on limitations in the discussion would be valuable. Perhaps the main unresolved point I see is the lack of explanation for why genetically identical flies, both wild type (control) and sleep mutants, sleep for such different durations (the within-group variation). In general, our scientific approaches and techniques almost solely rely on comparing group averages. This blinds us to important, meaningful biological and behavioral differences. The author's streamlined discussion leaves little room for the reader to imagine where this work can go next.

The paper is a meaningful advance. And linking the pressure to sleep to electron flow within mitochondria a major insight. But we still don't seem to know why different individuals need to sleep how much, at different times. Coming from this group, a note helping readers and the field in general to appreciate the need for explanatory processes/mechanisms operating at the individual level could stimulate future work around the bioenergetic mechanisms of sleep and behavior regulation?

2. The abstract only mentions "oxidative stress" as the potential culprit for the pressure to sleep. However, several pieces of the author's data, as well as some of their unpublished/preprint work, instead points to "reductive stress" as the key element.

The current version of the abstract reads "[...] predisposes them to heightened oxidative stress⁷. Consistent with this view, uncoupling electron flux from ATP synthesis¹⁵ relieves the pressure to sleep, while exacerbating mismatches between electron supply and ATP demand (by powering ATP synthesis with a light-driven proton pump¹⁶) promotes sleep." As the authors know, the most direct consequence of an imbalance between between electron flow and ATP synthesis is reductive stress. Acting as the primary driver, reductive stress then leads to oxidative stress, in some cases. Work by the Mootha lab has documented the mitochondrial control of reductive stress quite well, and established it, rather than oxidative stress, as the primary driver of transcriptional responses, cellular and animal phenotypes/behaviors (e.g., PMID: 32463360). I encourage the authors to either replace oxidative stress with reductive stress so as not to mislead the next decade of science in this field; or to use both terms, adding a sentence or two in the introduction to explain the concept of reductive stress, and for better up their upcoming papers.

3. The discussion is compelling and somewhat tantalizing, linking the emergence of the pressure to sleep with oxygen and nervous systems, and the amount of sleep needed with the rate of electron flow through mitochondria. This aligns directly with the rate of living hypothesis linking the rate of electron flow with lifespan. However, this hypothesis has flaws, highlighted by two main factors.

The first is that there are always outliers and broad scatter in plots of energy expenditure (O₂ consumption) and behaviors/lifespan. There is something fundamentally true about electron flow and behaviors, but electron flow alone cannot be the full explanation. The second is that mitochondrial respiratory chain defects (as in human mitochondrial diseases, for example) alone can drive behaviors. This happens without major alterations in the total rate of electron flow through mitochondria, or energy expenditure, although some evidence points to hypermetabolism in affected patients (PMID: 36635485). This state appears to result in fatigue and the pressure to sleep in humans (PMID: 26031904). So we learn from this that molecular defects in mitochondria, which increase the resistance to electron flow and cause reductive stress (PMID: 33463549), may be an important driver of signaling and cell/organismal behavior, without a necessary change in total

electron flow.

A recent proposal to resolve integrate both factors is the recently proposed energy resistance principle (https://osf.io/preprints/osf/hgnmj_v2). Borrowing from the power law in electrical engineering, this principle relates both electron flow (current) and the need for energy (power) to derive a biologically meaningful energy resistance term. This framework and the concept of energy resistance (éR) may represent the next regime that is in fact monitored by biological organisms, to adjust their behaviors including sleep. If so, energy resistance may hold the maximal explanatory power and experimental tractability in addressing the important problems addressed in this paper and in the author's ongoing work.

I am grateful for the opportunity to review and comment on this important work.

Response to Reviewers of Nature Manuscript 2024-02-03788

We thank the referees for their comments and suggestions. All significant changes in the revised manuscript (other than small stylistic alterations and changes to figure and reference numbers) are highlighted in red.

Referee 1

In this manuscript, Sarnataro and colleagues from the Miesenböck laboratory use *D. melanogaster* as a model to explore the biological basis of sleep pressure. Using a range of tools from single-cell transcriptomics to in vivo imaging and sleep measurements they show that mitochondrial dynamics and bioenergetics in dorsal fan-shaped body (dFBNs) are involved in determining the sleep pressure after deprivation.

This is an exciting claim. But I have two major concerns – first, for the non-sleep or fly experts (to which I belong), it is not immediately clear how this study fundamentally moves beyond the Miesenböck's prior paper entitled “A potassium channel β -subunit couples mitochondrial electron transport to sleep” from 2019, which uses similar tools in similar contexts, and in my reading already supports a view that there is an immediate link between mitochondrial respiration and sleep. I understand that the new manuscript (together with several others that are submitted in parallel) adds a lot more detail and claims to have discovered the biological substrate of sleep pressure (namely the ATP level and the corresponding redox equilibria of the respiratory chain), further linking this to the mitochondrial fission/fusion balance in specific neurons. Eventually, this is an editorial decision, also based on considering the other submitted manuscripts by the same group. But I at least could not exactly discern how fundamentally novel the proposed concepts are, and I was surprised to see a whole segment in the Results (paragraph 2 first part of paragraph 3 on page 5) that in essence say that some key data were published prior or will be published elsewhere.

The focus of our earlier paper was the β -subunit of the voltage-gated potassium channel Shaker and its role in the regulation of sleep. The notion that the β -subunit responds to by-products of respiration rested on two pieces of evidence¹: the detection of elevated levels of reactive oxygen species (ROS) after sleep deprivation, and a reduction in baseline sleep following the expression of the alternative oxidase AOX in dFBNs. AOX provided the only prior experimental link to mitochondria, and ROS levels the only link to sleep pressure.

The conceptual advances of the present study are threefold. First, single-cell transcriptomics offers independent, hypothesis-agnostic support for the idea that sleep and aerobic metabolism are fundamentally connected. Second, the current manuscript discovers, through numerous measurements and/or perturbations (of sleep history, ATP levels, neuronal activity, protonmotive force, and mitochondrial dynamics), that an imbalance between electron supply and ATP demand in dFBN mitochondria dictates the need for sleep. The relationship between respiration and sleep pressure is bidirectional and causal: sleep deficits change mitochondrial structure and function; experimental interference with mitochondria changes the homeostatic response to sleep loss. None of these insights were (or could have been) anticipated by our earlier work. Third, the study defines the molecular events that couple mitochondrial electron flow and ATP synthesis to sleep-promoting electrical activity, and electrical activity back to energy metabolism. To

be self-contained, the exposition of this mechanism introduces players established in earlier¹ ($K_V\beta$) or parallel work² (lipid peroxidation products), and it briefly describes the mode of action and impact on sleep of key tools (AOX), even if they have been used before.

However, beyond this somewhat subjective assessment, I also think the manuscript makes many assumptions about the tools employed that need substantial further controls, as detailed below.

Specific concerns:

The authors use a plethora of tools to manipulate mitochondria and measure the effects on various aspects of sleep. My problem is that most of the tools are not verified in detail. I understand that the authors would argue that perhaps the multiple streams of evidence make their claims independent of any single tool performing exactly as assumed, but I still think this needs to be shown in the specific settings used here.

We have tried our best to verify directly as many of our tools and analyses as possible and describe these new experiments below. Our point-by-point response also refers to published evidence, where available, that tools perform as expected.

Examples are:

Fig.1 – The extent and specificity for mitochondrial genes of the transcriptome changes are impressive – but which part of this can be translated to the functionally relevant proteome of mitochondria during the relevant time period (which I assume is 12 hours). In other words, what changes in mitochondrial molecular composition actually result for the sleep-deprivation induced cellular transcriptome – or even in the natural sleep-wake cycle?

Two lines of evidence indicate that changes in protein levels follow the transcriptomic changes we observe. First, super-resolution imaging of fluorescently labelled mitochondria reveals increases in total mitochondrial volume (the product of organelle count and average volume) after sleep deprivation (Fig. 3b). This net increase in mitochondrial mass is a clear indication that upregulated genes are translated. Second, a method termed STaR (synaptic tagging with recombination³) allowed us to add an epitope tag to an endogenous synaptic protein (BRP) whose mRNA is downregulated in dFBNs after sleep loss. Fig. 6 of our companion paper⁴ shows a corresponding reduction in protein expression. Unfortunately, however, STaR currently works only with the active zone protein BRP³ and not with any mitochondrial proteins.

*Attempts to measure specific changes in the molecular composition of mitochondria are hampered by a lack of reagents beyond STaR: neither antibodies effective in *Drosophila* nor YFP-trap alleles of the Cambridge Protein Trap Consortium collection⁵ target the products of any of our upregulated transcripts. Even if suitable reagents were available, a substantial limitation of bulk immunohistochemistry or protein traps (which cell-specific recombination methods like STaR have been designed to overcome) is that all endogenous proteins are labelled. Quantifying protein levels accurately within a tiny minority of 15 dFBNs—against the ubiquitous fluorescence background of ~70,000 neurons per brain hemisphere⁶—is challenging, to put it mildly. In fact, no published single-cell transcriptome of a complex tissue (in *Drosophila* or any other species) has to our knowledge been compared with single-cell proteomics.*

The amount of input material needed for cell-specific bulk methods⁷ such as APEX—250 µg—would require that approximately 3 million flies are individually sleep-deprived and dissected by hand.

Fig 2b,c – The authors claim a “15-fold higher” ATP level in deprived neurons. I think this needs in situ calibration, which could be tricky.

We thank the reviewer for spotting the missing decimal point, and for suggesting an in situ calibration. We have constructed a standard curve by dialyzing dFBNs with three defined ATP concentrations in whole-cell patch-clamp experiments and revised our estimate: ATP levels rise from 1.08 to 1.29 mM following a night of sleep loss, an increase of ~1.2-fold rather than ~1.5-fold (the new Extended Data Fig. 4a).

Fig. 2e – Why is there no recovery after the illumination ceases – is this a channelrhodopsin that stays activated or is this a change in sleeping status?

The original Fig. 2e showed measurements with the first-generation ATP sensor ATeam. It is likely that the seemingly incomplete recovery was caused by uneven bleaching of fluorescence in the two channels. Replacing ATeam with the second-generation sensor iATPSnFR demonstrates that ATP levels recover fully under the same stimulation conditions (the new Fig. 2e).

Fig. 2g,h – Where is the mitochondrial channelrhodopsin verified in its specific action? I could only find a general reference in other biological contexts

The light-driven proton pump delta-rhodopsin was previously expressed in Drosophila and shown to increase ATP levels in dopaminergic neurons upon illumination⁸. We have verified the ability to power ATP synthesis by direct optical measurement of light-driven ATP rises in dFBNs in vivo (the new Extended Data Fig. 4d).

Fig 3a,b – I have serious doubts that with the indicated techniques and work flow the morphology of mitochondria can be quantitatively captured – I know there is a reference to this workflow, but in my reading this is not established for flies or the specific imaging conditions used here. To control this, the authors should compare their data to published EM data, but ideally perform their own ultrastructural analysis also of the altered mitochondria, to prove that their read-out at the light microscopy level is valid for baseline and for their manipulations. I have seen data (e.g. in Justs et al., J Neurosci 2022) that seem to indicate that compared to mitochondria in other fly neurons, the volume estimates given here might be inflated (suggesting that the diffraction limit still determines much of what is measured). Please also explain, why the dendrites are the key compartment here – as I assume, somatic, or at least axonal mitochondria are not captured.

We are grateful for the suggestion to compare our morphological metrics to electron microscopy data published as part of the hemibrain connectome⁹. Extended Data Fig. 5a, b reports the results of this comparison. While diffraction undoubtedly inflated our absolute mitochondrial size estimates, characteristic relative size differences between the mitochondria of two neuron types¹⁰ (dFBNs and uniglomerular olfactory PNs of the antennal lobes) were accurately resolved. These size differences are similar to those seen in dFBNs after sleep deprivation and recovery or after manipulations of the mitochondrial fission and fusion machineries (the new Extended Data Fig. 7).

We have replicated our original morphological analyses with an optical super-resolution method¹¹, again calibrated against the ultrastructural ground truth^{9,10} (Extended Data Fig. 5a, b). The results, reported in the revised Fig. 3a, b, corroborate our original conclusions (now Extended Data Fig. 5e–g).

We also attempted an EM analysis of dFBNs but found the sparseness of their processes in tissue to be an insurmountable barrier: dFBNs represent a mere 15 out of ~70,000 neurons per hemisphere⁶; their somata do not lie immediately next to each other; and their filigree arbors span sizeable volumes. After successfully decorating dFBNs with the help of a cell surface-anchored horseradish peroxidase¹² (CD2::HRP) and collecting exploratory TEM sections at 4 μm spacing through five DAB-stained and three control blocks of whole brains (Fig. Aa), we found what looked like labelled profiles in only two of hundreds of high-magnification fields of view we examined (Fig. Ab). The labelled profiles happened to lack mitochondria.

Fig. A | Transmission electron microscopy.

a, Example brightfield images at different focal depths of the brain of a fly expressing *R23E10-GAL4-driven CD2::HRP*. The DAB signal was developed (ImmPACT DAB Peroxidase Substrate kit) after tyramide amplification (VECTASTAIN ABC-HRP Kit Peroxidase). Axons, dendrites, and somata of dFBNs are clearly visible. **b**, Example transmission electron micrograph of a section through the central complex of a DAB-stained block. Arrows identify structures potentially representing cross-sections of labelled dFBN neurites.

It is impossible to imagine how this approach could yield sufficient data for statistical inference, let alone how it could be applied to multiple individuals in the seven different experimental cohorts in Fig. 3. Encodable EM markers like *CD2::HRP* (Fig. A) have been available in *Drosophila* for two decades^{12,13} but found little use beyond the initial proof-of-principle studies of favourable anatomical structures, underscoring the immense difficulty of visualizing specific, rare neuron types in the fly brain without volume reconstruction.

We have added a clarifying statement to Methods that dendrites were chosen as a favourable compartment for analysis because their branches (and the mitochondria within them) are reasonably well separated along all axes.

Fig. 3d – What is the ground truth here – I cannot determine the specificity of DRP translocation in these highly processed images.

For localizing Drp1, we labelled the mitochondria of dFBNs with mito-GFP in Drp1::FLAG-FLAsH-HA flies¹⁴, whose genomic Drp1 coding sequence is fused in frame with a FLAG-FLAsH-HA tag. Because the endogenous Drp1 locus is modified, immunostaining with an anti-FLAG antibody labels every cell in the body. To restrict our analysis to dFBNs, we identified the somata of these neurons in the mito-GFP channel, which was then automatically thresholded, despeckled, and binarized. The mitochondria-associated fraction of Drp1 in dFBNs was quantified in summed z-stacks as the proportion of red-fluorescent pixels within the somatic volume that colocalized with mitochondrial objects. The original image was unprocessed, with the exception of the overlay of a semi-opaque mask defining the mitochondrial contours; the revised Fig. 3d separates the two channels and displays raw images of mito-GFP and Drp1-FLAG immunofluorescence in single focal planes.

Fig. 3e – Also here, EM evidence will be needed, and it needs to be shown that the contact sensor itself does not induce aggregation (it seems there is an effect outside the dFNB dendrites, see cluster to the lower right).

Please see above for an explanation of why the collection of EM evidence from specific rare neurons in the fly brain is incomparably more difficult than are EM studies of cultured cells or mammalian neurons; the technical obstacles are prohibitive.

Like our analyses of mitochondrial size and shape, the quantification of mitochondria–ER contact sites (MERCs) by light microscopy draws comparisons between dFBNs in rested and sleep-deprived brains. If the contact sensor¹⁵ SPLICS_{short} itself induced the formation of MERCs, we would expect to see no sleep history-dependent differences in contact site number, as the sensor was present in equal amounts in all conditions. This is clearly not the case (Fig. 3d).

The expression of SPLICS_{short} in HeLa cells does not change Ca²⁺ fluxes from one organelle to another (a very sensitive functional index of contact site formation¹⁶); SPLICS puncta form and dissolve dynamically in HeLa cells, zebrafish sensory neurons, and mouse hippocampal neurons¹⁶; the number of detected puncta does not vary with the expression level of the sensor¹⁵; mitochondrial morphology is unaltered by its presence¹⁵; and the expression of SPLICS in dFBNs has no impact on sleep in our hands (the new Extended Data Fig. 5h).

MERCs can be detected throughout dFBNs, including in the dendritic shaft that is visible to the lower right of Fig. 3d. For quantification we concentrated on the dendritic fields of dFBNs (the volume projected onto the dotted area in Fig. 3d), where individual SPLICS puncta could be resolved and counted.

Fig. 3f – The provided image does not seem representative of (or even on the same scale as) the corresponding graphs

The images are representative and displayed at the correct scale. The graph plots the average red-to-green ratios of all pixels in dendritic regions of interest, whereas the intensity scale next to the images spans the full range of red-to-green ratios measured in single pixels.

Fig. 4a,b: It needs to be shown (a) how the genetic manipulations affect protein levels in dFNBs and (b) how that affects mitochondrial morphology (this is the domain of EM or superresolution imaging, not confocal).

The new Extended Data Fig. 7 shows, using super-resolution imaging¹¹, that genetic manipulations of the mitochondrial fission and fusion machineries have the expected morphological consequences.

We have verified that epitope-tagged overexpression constructs of Drp1, Marf, and Opa1 are present in dFNBs, but it is unfortunately not technically feasible to quantify endogenous protein levels in a small group of neurons in the intact brain, as would be required to demonstrate the effectiveness of RNAi knockdowns. The fact that multiple independent RNAi transgenes have very similar effects, however, offers strong reassurance (Extended Data Fig. 8a–e).

Minor comments

Overall, the referencing is sometimes very broad and should be limited to essential and primary literature, not general reviews, see p. 6 Ref 14-16.

We have prioritized citations of original research as far as possible. However, a reference count limit made it necessary to refer to a small number of reviews in order to cover vast primary literatures on hypotheses about the function of sleep, brain evolution, the geochemistry of Earth's history, or the interplay between mitochondrial morphology and function.

The discussion is interesting, but perhaps a bit remote from the data – and should at least discuss relevant recent papers such as Haynes et al., Nat Neurosci 2024 (which might have appeared only after the authors completed this manuscript), and perhaps also discuss the question whether there are any hints in other organisms suggesting a similar basis for sleep pressure.

The paper by Haynes and colleagues¹⁷ appeared online just as we submitted our original manuscript; we refer to it prominently in the revised text.

The discussion highlights correlations between mammalian metabolic rate, sleep amount, and life span¹⁸. These correlations appear rooted in body size-dependent differences in O₂ supply¹⁹ and thus point to mitochondrial respiration as the underlying cause, a notion supported by our own unpublished experimental evidence in the mouse.

Referee 2

The manuscript “Mitochondrial origins of the pressure to sleep” by Sarnataro et al. claims to demonstrate a central role of mitochondrial aerobic metabolism in generating the pressure to sleep. The hypothesis is novel and interesting and the authors bring to bear a slew of approaches and novel data to support the hypothesis. This work builds from the prior landmark paper (Kempf et al, 2019) showing that the buildup of mitochondrial-driven ROS activates sleep via the redox state of the K channel subunit Hyperkinetic. They extend these

findings here by demonstrating wake-driven changes in mitochondrial gene transcripts and ROS driven mitochondrial fragmentation. A nice set of genetic/optogenetic perturbation experiments demonstrate the interplay of electron flux and ATP synthesis/levels is critical to dFBN activity in the control of sleep. The strength of the paper is in the elucidation of a molecular and cellular pathway linking energy availability and mitochondrial metabolism to the control of neurons that regulate sleep.

Major Comments:

1. While the paper makes use of many sophisticated techniques to substantiate their model, it is not clear how this paper on its own substantially advances the model originally put forth in Kempf (see Fig. 2 of that paper)

The focus of our earlier paper¹ was the β -subunit of the voltage-gated potassium channel Shaker. Mitochondria entered the frame when we asked whether the β -subunit is a redox sensor; the respiratory chain is of course the primary site of cellular redox chemistry. The expression of the alternative oxidase AOX provided the only direct experimental link between sleep and mitochondria in earlier work¹.

The conceptual advances of the present study are threefold. First, single-cell transcriptomics offers independent, hypothesis-agnostic support for the idea that sleep and aerobic metabolism are fundamentally connected. Second, as the reviewer kindly notes, the current manuscript uses numerous new measurements and perturbations of sleep history, ATP levels, neuronal activity, protonmotive force, and mitochondrial dynamics to show that an imbalance between electron supply and ATP demand in dFBN mitochondria dictates the need for sleep. The relationship between respiration and sleep pressure is bidirectional and causal: sleep deficits change mitochondrial structure and function; experimental interference with mitochondria changes the homeostatic response to sleep loss. None of these insights were (or could have been) anticipated by our earlier work. Third, the study defines the molecular events that couple mitochondrial electron flow and ATP synthesis to sleep-promoting electrical activity, and electrical activity back to energy metabolism. We return to these events in our response to the Referee's point 4, below.

2. The paper is predicated on studies of the dFBN in sleep regulation. The authors in this manuscript make scant reference to two peer-reviewed studies (from the Joiner and Dissel groups) that have strongly questioned the preeminent role of the dFBN. From those studies it is clear that the 23E10 driver used widely in this study also drives expression in other potent sleep regulatory “bowtie” neurons. Thus the genetic/RNAi experiments (Fig. 2 and 4) using this driver ought to use a reagent that excludes a potential function in the “bowtie” neurons.

Stéphane Dissel's and our groups have gathered independent, concordant evidence that forcefully and directly refutes the challenge²⁰ to the sleep-regulatory role of dFBNs. These experiments, which are presented in full in our companion submission⁴ (available on BioRxiv) and a preprint²¹ by Jones et al., rely on a collection of split-GAL4 lines without expression in 'bowtie' VNC-SP neurons of the ventral nerve cord²², a suite of positive and negative regulators of neuronal activity or synaptic transmission, and measurements of baseline and rebound sleep.

In the current study, we have reproduced all behavioural results of the original Fig. 2 and Fig. 4 with the intersectional driver $R23E10 \cap VGlut-GAL4$, which targets glutamatergic

dFBNs of the central brain⁴ (the ‘bona fide dFBNs’ of Fig. 1) but excludes the cholinergic bowtie neurons which are present in the R23E10-GAL4 line. The results are identical to those obtained with R23E10-GAL4 (now Extended Data Fig. 4c–f and Extended Data Fig. 8).

Anissa Kempf has in her laboratory in Basel repeated all behavioural experiments of her original paper¹ with the intersectional driver²¹ R23E10 \cap R84C10-GAL4 and mapped the effects on sleep to dFBNs (please see the next paragraph).

3. A major component of their model is the behavioral function of Hk in the dFBNs which was questioned recently. It would be good if this could be confirmed, addressed or refuted.

The behavioural function of Hk in dFBNs was confirmed by us, using the R23E10-GAL4 driver² (Fig. 3b, c), and by Anissa Kempf, using the intersectional line²¹ R23E10 \cap R84C10-GAL4 and two independent HK^{RNAi} transgenes. Anissa’s results will

be available on BioRxiv soon. Fig. B previews the Hk^{RNAi} data.

4. Haynes et al has shown a role for the mitochondrial fission protein Drp1 in sleep regulation—this paper extends that work to show its function in the dFBN (although with the caveat listed above). Nonetheless, it was difficult to discern the changes in mitochondrial morphology in Fig. 3. I wonder whether such changes require ultrastructural analyses to be more definitive. It is also not clear mechanistically how these changes are linked to cellular physiology important for sleep regulation.

The paper by Haynes and colleagues¹⁷ appeared online just as we submitted our original manuscript; we refer to it prominently in the revised text.

We have replicated our original morphological analyses with an optical super-resolution method¹¹, which we calibrated against the ultrastructural ground truth^{9,10} (Extended Data Fig. 5a, b). Characteristic size differences between the mitochondria of two neuron types (dFBNs and uniglomerular olfactory projection neurons of the antennal lobes) were accurately resolved. These size differences are similar to those seen in dFBNs after sleep deprivation and recovery or after manipulations of the mitochondrial fission and fusion machineries (the new Extended Data Fig. 7). These results, reported in the revised Fig. 3a, b, corroborate our original conclusions (now Extended Data Fig. 5e–g).

We also attempted an EM analysis of dFBNs but found the sparseness of their processes in tissue to be an insurmountable barrier: dFBNs represent a mere 15 out of ~70,000 neurons per hemisphere⁶; their somata do not lie immediately next to each other; and their filigree arbors span sizeable volumes. After successfully decorating dFBNs with the help

of a cell surface-anchored horseradish peroxidase¹² (CD2::HRP) and collecting exploratory TEM sections at 4 μm spacing through five DAB-stained and three control blocks of whole brains (please see Fig. A in our response to Referee 1), we found what looked like labelled profiles in only two of hundreds of high-magnification fields of view we examined (Fig. Ab). The labelled profiles happened to lack mitochondria.

It is impossible to imagine how this approach could yield sufficient data for statistical inference, let alone how it could be applied to multiple individuals in the seven different experimental cohorts in Fig. 3. Encodable EM markers like CD2::HRP (Fig. A) have been available in Drosophila for two decades^{12,13} but found little use beyond the initial proof-of-principle studies of favourable anatomical structures, underscoring the immense difficulty of visualizing specific, rare neuron types in the fly brain without volume reconstruction.

Central to the connection between mitochondrial morphology, cellular biophysics, and sleep is a homeostatic feedback mechanism controlled by the balance of electron supply and ATP demand. The voltage-gated potassium channel Shaker senses this balance indirectly and couples it to the neurophysiology of sleep control. We show that dFBNs experience low ATP demand during extended wakefulness because arousing dopamine inhibits spiking (Fig. 2). This results in a mitochondrial electron surplus, the diversion of excess electrons into superoxide, and the peroxidation of membrane lipids². dFBNs respond i) by increasing their excitability (via the reduction of lipid peroxidation-derived carbonyls by Hk, which progressively converts the Hk pool to the NADP⁺-bound form²) and ii) by reducing electron flow (via mitochondrial fission, Fig. 3). Inducing fission artificially, as we have done in Fig. 4, reduces the electron leak and attenuates excitability and the pressure to sleep accordingly; inducing fusion has the opposite effects. The system naturally returns to baseline during recovery sleep, when i) the increased electrical activity^{4,23} of dFBNs consumes ATP (Fig. 2) and releases² NADP⁺ from Hk and ii) mitochondria fuse to meet the heightened demand for ATP (Fig. 3). This closes the feedback loop.

Referee 3

The manuscript by Sarnataro and colleagues presents a compelling set of experiments, in Drosophila, linking different aspects of mitochondrial biology with sleep regulation. The authors first use single-cell RNA sequencing to identify a population of dFBNs cells responsive to sleep deprivation (and/or stress). They then examine how manipulating different aspects of mitochondrial bioenergetics in this cell population influence sleep behavior on aggregate fly populations, showing that manipulation of mitochondrial dynamics genes influence sleep behavior in directions predicted by the model. The work appears to be of high quality and the question is of high importance. Some limitations of the work and opportunities to increase the impact of the manuscript are listed below.

In the scRNAseq data, there was a large difference in the number of dFBNs between the two groups (237 vs 86). Why is this? This potentially undermines the power and introduced the potential of a skewed population of cells being studied between the groups. How can the authors rule out this possibility?

The reason is technical: although roughly equally sized groups of flies entered the scRNAseq pipeline, many more flies were lost during mechanical sleep deprivation than

during unperturbed sleep. We emphasize that these losses are a side effect of the method used to prevent sleep, not a lethal consequence of sleep deprivation itself. To deprive flies of sleep, individual animals were placed in small glass tubes with a food plug on one end. The tubes were slowly tilted on a spring-loaded platform, which was allowed to snap back abruptly to its original position once every 10 s. At the end of the experiment, a not insignificant number of flies were found bogged down in food or crushed by plugs that had been shaken loose.

We have confirmed by resampling that reducing the number of transcriptomes of rested dFBNs to that of sleep-deprived dFBNs has no impact on our conclusions (the new Extended Data Fig. 2e).

Given the different distribution of differentially expressed genes in Figure 1b, it may be useful to include non-significant datapoints (the bottom of the volcano). Having the x axis go from -1.5 to +1.5 would also help readers appreciate relative differences between Up and Down regulated groups of genes. This also applies to Extended Data figures.

Gray symbols at the bottom of volcano plots indicate non-significant data points; the only data points omitted from the plots were those with P-values of 1.00000000 (rounded to the rigidly set decimal scale of the Seurat v4.1 package²⁴), which form an obtrusive line at $\log_{10}(P) = 0$. These data points are, however, included in the Source Data file.

No data points fall outside the current x-axis limits, which were chosen to avoid excessive white space.

The authors report that compared to rested, SD leads to a fifteen (15)-fold higher ATP levels. ATP levels tend to be strongly controlled and likely never vary that much in a living context. The datapoints also don't seem to show a 15-fold difference. Is this a mistake? Can the author clarify.

We thank the reviewer for spotting the missing decimal point. We have constructed a standard curve by dialyzing dFBNs with three defined ATP concentrations in whole-cell patch-clamp experiments and revised our estimate: ATP levels rise from 1.08 to 1.29 mM following a night of sleep loss, an increase of ~1.2-fold rather than 1.5-fold (the new Extended Data Fig. 4a).

Minor related point: if the points were semi-transparent the reader could appreciate the density of datapoints within the distribution, which is obscured in the current version of the figures.

We have reduced the size of all data points to make their distributions clearer.

The authors should provide data confirming the extent to which their cell-type specific expression mutants are in fact cell type specific. How much leak is there outside the target cell type?

We use a binary expression system (GAL4-UAS) to target genetic manipulations to specific cell types. Specificity could be compromised either by leaky expression of a UAS transgene (that is, expression in the absence of a GAL4 driver), or by expression of GAL4 outside the cells of interest.

We eliminate the risk of leaky UAS transgene expression in two ways: through the inclusion of controls carrying the transgene but lacking a GAL4 driver in all experiments,

and through the use of multiple independent transgene insertions. The paper exhausts every possibility in this regard; we use 3 independent transgenes for the overexpression of *Drp1*, 2 independent transgenes for the overexpression of *Marf*, 7 independent RNAi transgenes for interference with the expression of *Opal*, 5 independent RNAi transgenes directed against *Marf*, and 2 independent RNAi transgenes directed against *Mitoguardin*. The behavioural phenotypes are consistent across the board.

To confirm driver specificity, we use two *GAL4* lines targeting *dFBNs* (*R23E10* \cap *VGlut-GAL4* and *R23E10-GAL4*) interchangeably, and we verify the absence of sleep phenotypes when transgene expression is directed to two different neuronal populations, projection neurons of the antennal lobe (*GH146-GAL4*) and Kenyon cells of the mushroom body (*OK107-GAL4*) (Extended Data Fig. 10).

As presented, the mitochondrial morphology analysis is weak. The data for mitochondrial size and morphology appears to have been done using fairly low magnification. At low resolution with light microscopy, juxtaposed mitochondria (either post fission, or bound by inter-mitochondrial junctions, for example) will appear “fused” or longer than they actually are. The morphological alterations would be best demonstrated by transmission electron microscopy, or even more compellingly with serial scanning EM (FIB-SEM, SBF-SEM) where the morphology of mitochondria can be accurately determined.

We have replicated our original morphological analysis with an optical super-resolution method¹¹, which we calibrated against the ultrastructural ground truth^{9,10} (Extended Data Fig. 5a, b). Characteristic size differences between the mitochondria of two neuron types (*dFBNs* and uniglomerular olfactory projection neurons of the antennal lobes) were accurately resolved. These size differences are similar to those seen in *dFBNs* after sleep deprivation and recovery or after manipulations of the mitochondrial fission and fusion machineries (the new Extended Data Fig. 7). These results, reported in the revised Fig. 3a, b, corroborate our original conclusions (now Extended Data Fig. 5e–g).

We also attempted an EM analysis of *dFBNs* but found the sparseness of their processes in tissue to be an insurmountable barrier. *dFBNs* represent a mere 15 out of ~70,000 neurons per hemisphere⁶; their somata do not lie immediately next to each other; and their filigree arbors span sizeable volumes. After successfully decorating *dFBNs* with the help of a cell surface-anchored horseradish peroxidase¹² (*CD2::HRP*) and collecting exploratory TEM sections at 4 μ m spacing through five DAB-stained and three control blocks of whole brains (please see Fig. A in our response to Referee 1), we found what looked like labelled profiles in only two of hundreds of high-magnification fields of view we examined (Fig. Ab). The labelled profiles happened to lack mitochondria.

It is impossible to imagine how this approach could yield sufficient data for statistical inference, let alone how it could be applied to multiple individuals in the seven different experimental cohorts in Fig. 3. Encodable EM markers like *CD2::HRP* (Fig. A) have been available in *Drosophila* for two decades^{12,13} but found little use beyond the initial proof-of-principle studies of favourable anatomical structures, underscoring the immense difficulty of visualizing specific, rare neuron types in the fly brain without volume reconstruction.

The model presented suggests that sleep duration is influenced by the balance of mitochondrial fusion and fission. If this is correct and generalizable in *Drosophila*, as suggested by the experimental manipulations of the fission/fusion machinery, then this should

explain why some flies sleep more while some sleep less. In the data from Figure 4b, it is clear that some control R23E10 flies are short sleepers (<10h per day) while others are long sleepers (>20h). If the author's model is correct and sleep behavior is under the control of mitochondria, it should be possible to predict the behavior of each fly, or of groups of very short, short, average, long, and very long sleeping flies, based on gene expression or some other measure of the mitochondrial system. Can the authors address the relevance of their model related to inter-individual differences within one or more populations of flies? If they could address this point in both reference and, for example, Opa1-RNAi flies, this would be very compelling and would increase the impact of the paper. Specifically, what is the proportion of variance (r^2 on a regression analysis) in sleep behavior that is accounted for by expression of fission or fusion genes, or the ratio of fission>fusion genes? Finding a positive result would increase the likelihood that this mechanism may have relevance to understand inter-individual differences in sleep in other animals. Finding a negative result (hypothesis true across groups, but not between individuals of a population) would not necessarily invalidate the relevance of the model, but would force an additional layer of interpretation that also could ultimately improve the paper.

We describe a bidirectional, causal relationship between mitochondrial dynamics and the feedback regulation of sleep. Because mitochondria and sleep are both dynamic, experimental interference with mitochondria-shaping proteins or sleep is needed to 'clamp' one variable and quantify its impact on the other. Relating an individual's unperturbed sleep history to a snapshot of mitochondrial structure or gene expression in dFBNs is problematic because the key variable of momentary sleep pressure cannot be determined retrospectively. If a fly spent many hours asleep before the mitochondrial snapshot was taken, is this an indication that sleep pressure remained persistently high, or is it a sign of effective homeostatic compensation that lowered sleep need at the moment of analysis? Only future sleep could tell the difference, but this information is unfortunately unattainable after a fly has been dissected.

As far as gene expression is concerned, we cannot measure transcript levels in dFBNs of individual flies but note that genes encoding mitochondrial fission and fusion proteins are not differentially expressed in our transcriptomic data. This observation suggests that control over the mitochondria-shaping machinery is exerted primarily by proteolysis, phosphorylation, or other posttranscriptional modifications.

I am not an electrophysiologist and have little understanding of the constraints associated with these experiments. That said, it seems that one additional approach to test the question above may be to relate the electrophysiological measures in Figure 4D with sleep behavior in individual flies. First establish how long a fly sleeps per day then measure electrophysiology on the same fly, and correlate these parameters to see how much variance is shared between these measurements. Are the electrophysiological properties in a fly sufficiently dominant and causal that they can be used to predict sleep behavior? Here the potential for bias is high, so these experiments should be done blinded where the person determining sleep duration creates a code (Fly A, B, C, D, ...) so the sleep duration data cannot include the electrophysiological recordings.

The fundamental impossibility of inferring the amount of 'unclamped' sleep pressure retrospectively also applies here. Under clamped conditions (that is, after sleep deprivation), electrophysiological recordings²³ from dFBNs and optical imaging of slow-

wave activity⁴ have connected changes in the biophysics of these neurons to the experimentally controlled sleep history of their owners.

Is the approach to sleep deprive fly also “stressful”? How can the effects of sleep deprivation disentangled from those of stress responses or adaptation?

Attempts to distinguish the consequences of sleep deprivation from independent responses to the stimuli that are required to keep animals awake rely on the use of yoked controls²⁵ or genetic methods of sleep deprivation without mechanical agitation, using either mutants or the genetically targeted activation of wake-promoting neurons. We have used the latter approach, namely, sleep deprivation via activation of arousing dopaminergic neurons with the temperature-sensitive cation channel TrpA1. The new Extended Data Fig. 5c, d shows that genetic sleep deprivation elicits the same changes in mitochondrial morphology as does mechanical agitation.

In some behavioural experiments, we have used a method where mechanical displacements occur at random rather than regular intervals to guard against possible adaptation (Extended Data Fig. 8f), again with effects that are indistinguishable from those of our standard method (Fig. 4d).

References 2 and 5 are on BioRxiv. It would have helped to indicate as such, especially given that reference 5 is used to support some of this manuscript.

We regret this omission. Even though the citations were not available at the time of submission, we should have indicated that the manuscripts were due to appear on BioRxiv. Our apologies.

References

1. Kempf, A., Song, S. M., Talbot, C. B. & Miesenböck, G. A potassium channel β -subunit couples mitochondrial electron transport to sleep. *Nature* **568**, 230-234 (2019).
2. Rorsman, H. O. et al. Sleep pressure accumulates in a voltage-gated lipid peroxidation memory, *Nature*, in press (2025). (*BioRxiv* 2024.02.25.581768)
3. Chen, Y. et al. Cell-type-specific labeling of synapses in vivo through synaptic tagging with recombination. *Neuron* **81**, 280-293 (2014).
4. Hasenhuettl, P. S. et al. A half-centre oscillator encodes sleep pressure. *BioRxiv* 2024.02.23.581780 (2024).
5. Lowe, N. et al. Analysis of the expression patterns, subcellular localisations and interaction partners of Drosophila proteins using a pigP protein trap library. *Development* **141**, 3994-4005 (2014).
6. Hulse, B. K. et al. A connectome of the Drosophila central complex reveals network motifs suitable for flexible navigation and context-dependent action selection. *Elife* **10**, e66039 (2021).
7. Chen, C.-L. et al. Proteomic mapping in live Drosophila tissues using an engineered ascorbate peroxidase. *Proc Natl Acad Sci U S A* **112**, 12093-12098 (2015).

8. Imai, Y. et al. Light-driven activation of mitochondrial proton-motive force improves motor behaviors in a *Drosophila* model of Parkinson's disease. *Commun Biol* **2**, 424 (2019).
9. Scheffer, L. K. et al. A connectome and analysis of the adult *Drosophila* central brain. *Elife* **9**, e57443 (2020).
10. Rivlin, P. K. et al. Connectomic analysis of mitochondria in the central brain of *Drosophila*. *bioRxiv* 2024.04.21.590464 (2024).
11. Azuma, T. & Kei, T. Super-resolution spinning-disk confocal microscopy using optical photon reassignment. *Opt Express* **23**, 15003-15011 (2015).
12. Watts, R. J., Schuldiner, O., Perrino, J., Larsen, C. & Luo, L. Glia engulf degenerating axons during developmental axon pruning. *Curr Biol* **14**, 678-684 (2004).
13. Ng, J. et al. Genetically targeted 3D visualisation of *Drosophila* neurons under electron microscopy and X-ray microscopy using miniSOG. *Sci Rep* **6**, 38863 (2016).
14. Verstreken, P. et al. Synaptic mitochondria are critical for mobilization of reserve pool vesicles at *Drosophila* neuromuscular junctions. *Neuron* **47**, 365-378 (2005).
15. Cieri, D. et al. SPLICS: A split green fluorescent protein-based contact site sensor for narrow and wide heterotypic organelle juxtaposition. *Cell Death Differ* **25**, 1131-1145 (2018).
16. Vallese, F. et al. An expanded palette of improved SPLICS reporters detects multiple organelle contacts in vitro and in vivo. *Nat Commun* **11**, 6069 (2020).
17. Haynes, P. R. et al. A neuron-glia lipid metabolic cycle couples daily sleep to mitochondrial homeostasis. *Nat Neurosci* **27**, 666-678 (2024).
18. Zepelin, H. & Rechtschaffen, A. Mammalian sleep, longevity, and energy metabolism. *Brain Behav Evol* **10**, 425-470 (1974).
19. Savage, V. M. & West, G. B. A quantitative, theoretical framework for understanding mammalian sleep. *Proc Natl Acad Sci U S A* **104**, 1051-1056 (2007).
20. De, J., Wu, M., Lambatan, V., Hua, Y. & Joiner, W. J. Re-examining the role of the dorsal fan-shaped body in promoting sleep in *Drosophila*. *Curr Biol* **33**, 3660-3668.e4 (2023).
21. Jones, J. D. et al. The dorsal fan-shaped body is a neurochemically heterogeneous sleep-regulating center in *Drosophila*. *bioRxiv* 2024.04.10.588925 (2024).
22. Jones, J. D. et al. Regulation of sleep by cholinergic neurons located outside the central brain in *Drosophila*. *PLoS Biol* **21**, e3002012 (2023).
23. Donlea, J. M., Pimentel, D. & Miesenböck, G. Neuronal machinery of sleep homeostasis in *Drosophila*. *Neuron* **81**, 860-872 (2014).
24. Hao, Y. et al. Integrated analysis of multimodal single-cell data. *Cell* **184**, 3573-3587.e29 (2021).
25. Rechtschaffen, A., Gilliland, M. A., Bergmann, B. M. & Winter, J. B. Physiological correlates of prolonged sleep deprivation in rats. *Science* **221**, 182-184 (1983).

Response to Reviewers of Nature Manuscript 2024-02-03788A

We thank the referees for their comments and suggestions, which we found most helpful. It would be a joy to experience this standard of reviewing more often.

Referee 1

The authors have carefully and constructively addressed my concerns - thank you!

I appreciate their efforts to even address difficult challenges (such as the calibration of the ATP sensor), their explanation of unsurmountable technical problems (e.g. regarding my suggestions for EM analysis) and the clarification and correction of this that I either misunderstood or where I spotted minor mistakes.

So altogether, I am satisfied with the technical revisions made to address my concerns. For the question of novelty and the relationship to recent advances in the literature, I think the editorial team has to decide - but I would assume the invitation of a revision to the authors and a re-review to the referees, underscores the fact that editorially this manuscript is seen favorably.

I think it is an impressive piece of work that will stimulate the growing field that ascribes systemic effects to neuronal mitochondria. I would be excited to see this work in print!

Thank you.

Referee 2

The authors have made significant revisions addressing some concerns. The major issue that remains is the novelty of this paper relative to their prior publication Kempf et al. In the Fig. 2 legend of that paper the authors state, "When more electrons enter the transport chain than are needed for ATP synthesis, a backlog accumulates in the Q pool." This appears to be conceptually very close to the "imbalance" model that is presented in this paper, albeit with more experimental evidence .

The authors have also added important new data to strengthen the manuscript including adding refined split GAL4 drivers to exclude bowtie VNC-SP neurons in their key genetic experiments. It is noteworthy that some of the functional effects (e.g, Fig. 4C) are relatively modest with relatively large sample sizes. That being said, it is clear that the dFBNs do have a sleep regulatory role and that perturbations of mitochondrial fission and fusion do contribute to this function. In addition, the authors have gone to great lengths and made significant progress on substantiating the morphological changes using an optical super-resolution method. Collectively the mitochondrial fission/fusion role in sleep homeostasis is an advance, although with the caveat that Haynes et al had described the sleep function of the mitochondrial fission protein Drp1.

We are grateful for the referee's comments. We have previously stated the case for the advances the current paper makes over Kempf et al. (and also over the suggestion of a sleep-regulatory role of Drp1 in Haynes et al.) and will not reiterate them here. Comparisons of effect sizes obtained with different GAL4 drivers ought to be drawn with caution.

Referee 3

The manuscript is significantly improved and all major points of concern have been addressed.

Three suggestions:

1. A short, lucid paragraph on limitations in the discussion would be valuable. Perhaps the main unresolved point I see is the lack of explanation for why genetically identical flies, both wild type (control) and sleep mutants, sleep for such different durations (the within-group variation). In general, our scientific approaches and techniques almost solely rely on comparing group averages. This blinds us to important, meaningful biological and behavioral differences. The author's streamlined discussion leaves little room for the reader to imagine where this work can go next.

The paper is a meaningful advance. And linking the pressure to sleep to electron flow within mitochondria a major insight. But we still don't seem to know why different individuals need to sleep how much, at different times. Coming from this group, a note helping readers and the field in general to appreciate the need for explanatory processes/mechanisms operating at the individual level could stimulate future work around the bioenergetic mechanisms of sleep and behavior regulation?

We are grateful for this suggestion and have added a comment about potential sources of interindividual differences to the discussion. Variation in mitochondrial genomes is one, but certainly not the only, possible factor, as the expression of many behaviours differs among isogenic individuals, reflecting perhaps the stochasticity of neuronal development or function.

2. The abstract only mentions "oxidative stress" as the potential culprit for the pressure to sleep. However, several pieces of the author's data, as well as some of their unpublished/preprint work, instead points to "reductive stress" as the key element.

The current version of the abstract reads "[...] predisposes them to heightened oxidative stress⁷. Consistent with this view, uncoupling electron flux from ATP synthesis¹⁵ relieves the pressure to sleep, while exacerbating mismatches between electron supply and ATP demand (by powering ATP synthesis with a light-driven proton pump¹⁶) promotes sleep." As the authors know, the most direct consequence of an imbalance between between electron flow and ATP synthesis is reductive stress. Acting as the primary driver, reductive stress then leads to oxidative stress, in some cases. Work by the Mootha lab has documented the mitochondrial control of reductive stress quite well, and established it, rather than oxidative stress, as the primary driver of transcriptional responses, cellular and animal phenotypes/behaviors (e.g., PMID: 32463360). I encourage the authors to either replace oxidative stress with reductive stress so as not to mislead the next decade of science in this field; or to use both terms, adding a sentence or two in the introduction to explain the concept of reductive stress, and for better up their upcoming papers.

We agree with the referee's comment about the primary role of reductive stress, which then leads to oxidative stress. The revised abstract avoids either term (in order not to confuse the uninitiated, which are likely to be unfamiliar with the concept of reductive stress) and instead speaks of an augmented electron leak.

3. The discussion is compelling and somewhat tantalizing, linking the emergence of the pressure to sleep with oxygen and nervous systems, and the amount of sleep needed with the rate of electron flow through mitochondria. This aligns directly with the rate of living hypothesis linking the rate of electron flow with lifespan. However, this hypothesis has flaws, highlighted by two main factors.

The first is that there are always outliers and broad scatter in plots of energy expenditure (O₂ consumption) and behaviors/lifespan. There is something fundamentally true about electron flow and behaviors, but electron flow alone cannot be the full explanation. The second is that mitochondrial respiratory chain defects (as in human mitochondrial diseases, for example) alone can drive behaviors. This happens without major alterations in the total rate of electron flow through mitochondria, or energy expenditure, although some evidence points to hypermetabolism in affected patients (PMID: 36635485). This state appears to result in fatigue and the pressure to sleep in humans (PMID: 26031904). So we learn from this that molecular defects in mitochondria, which increase the resistance to electron flow and cause reductive stress (PMID: 33463549), may be an important driver of signaling and cell/organismal behavior, without a necessary change in total electron flow.

A recent proposal to resolve integrate both factors is the recently proposed energy resistance principle (https://osf.io/preprints/osf/hgnmj_v2). Borrowing from the power law in electrical engineering, this principle relates both electron flow (current) and the need for energy (power) to derive a biologically meaningful energy resistance term. This framework and the concept of energy resistance (ϵR) may represent the next regime that is in fact monitored by biological organisms, to adjust their behaviors including sleep. If so, energy resistance may hold the maximal explanatory power and experimental tractability in addressing the important problems addressed in this paper and in the author's ongoing work.

I am grateful for the opportunity to review and comment on this important work.

Again, we share the referee's point of view. We hope that our revised abstract and discussion make clear that we see the mitochondrial electron leak as key. The size of the leak is determined not only by the magnitude of the electrical current flowing through the respiratory chain but also the chain's resistance.